# On the Utility of Existing Fine-Tuned Models on Data-Scarce Domains

**Md Ibrahim Ibne Alam**  *alamm4@rpi.edu*
*Department of Electrical, Computer, and Systems Engineering*
*Rensselaer Polytechnic Institute*

**Parikshit Ram**  *parikshit.Ram@ibm.com*
*IBM*

**Soham Dan**  *sdan021@gmail.com*
*Microsoft*

**Horst Samulowitz**  *samulowitz@us.ibm.com*
*IBM*

**Koushik Kar**  *koushik@ecse.rpi.edu*
*Department of Electrical, Computer, and Systems Engineering*
*Rensselaer Polytechnic Institute*

**Reviewed on OpenReview:** *https://openreview.net/forum?id=kY2fKLOGkI*

## Abstract

Large Language Models (LLMs) have been observed to perform well on a wide range of downstream tasks when fine-tuned on domain-specific data. However, such data may not be readily available in many applications, motivating zero-shot or few-shot approaches using existing *domain or task adjacent (fine-tuned) models*, which we call DAFT. While several fine-tuned models for various tasks are available, finding one appropriate DAFT model for a given task is often not straight forward. In this paper, we explore different utilization techniques of these existing DAFT models for data-scarce problems, i.e., tasks for which data is not available or limited. We observe that for zero-shot problems, ensembling of DAFT models provides an accuracy performance close to that of the single best model. With few-shot problems (few data from target domain available), this performance can be improved further by picking or putting more weights to the DAFT models that are expected to perform better on the target task.

## 1 Introduction

Pre-trained Large Language Models (LLMs) are used for different downstream tasks. Usually, for LLMs that are not directly suitable for downstream tasks, a task-specific header layer, e.g., for classification, is needed at the output. *We define the* base model *as the pre-trained LLM with a task-specific (un-tuned) header layer.* As an example, BERT models (Kenton & Toutanova, 2019) were trained for sentence completion tasks. In order to perform sentiment classification, we need to add a header layer with output dimension matching the number of classes. Usually, *base models* do not need much training to perform well, because the pre-trained LLMs are already trained on a large corpus of data.

To perform a task with base models, we can fine-tune a base model with task-specific training data, and then use the fine-tuned model. This fine-tuning of model is computationally much less extensive compared to the initial training phase of the LLMs. However, it is often the case that appropriate training data for fine-tuning is not readily or abundantly available, i.e., the domain is data-scarce. Moreover, even if one

does have the data to fine-tune, one may not have the computational resources, or the time required, to fine-tune. It is worth noting that fine-tuning can be computation, memory and time intensive, even if very few iterations are needed to attain good performance. Hence, we compare different fine-tuning solutions with some alternatives that bypass the challenges of fine-tuning and leverage the large number of fine-tuned LLMs already available (in curated repositories like Hugging-face (HF, 2024c; Kim, 2023)). It is important to acknowledge the scale of fine-tuned models being created; for instance, *just three days after LLaMA-3 was released, more than 1,000 fine-tuned LLMs of LLaMA-3* were publicly available on Hugging-face (HF, 2024b).

Research Question: Can we use these plethora of fine-tuned LLMs for a data-scarce task?

Here, we investigate the potential of existing and publicly available fine-tuned models (e.g., including those fine-tuned via LoRA (Sheng et al., 2023) and PEFT (Li & Liang, 2021)) by leveraging them to perform different tasks. To use a fine-tuned model on a task, e.g., sentiment classification, we need that model to be *domain or task adjacent* (discussed in Section 3). We denote these **D**omain **A**djacent **F**ine-**T**uned models as DAFT. One main property of DAFT is that it can be used without any further fine-tuning (no extra computation) in a zero-shot manner (no data adaptation). Also, inference with DAFT does not require large computational resources when compared to performing fine-tuning.

The availability of DAFT models combined with their ease of use, make them very appealing to use for different tasks. However, the performance of DAFT models on the target domain data (denoted as $D_T$) can be poor depending on the fine-tuning dataset of the respective DAFT models. In this work, we *systematically study four different ways (Fig. 1) of leveraging the DAFT models*, and demonstrate how it is possible to get strong performance with low compute adaptation, demonstrating the utility of these community gener-

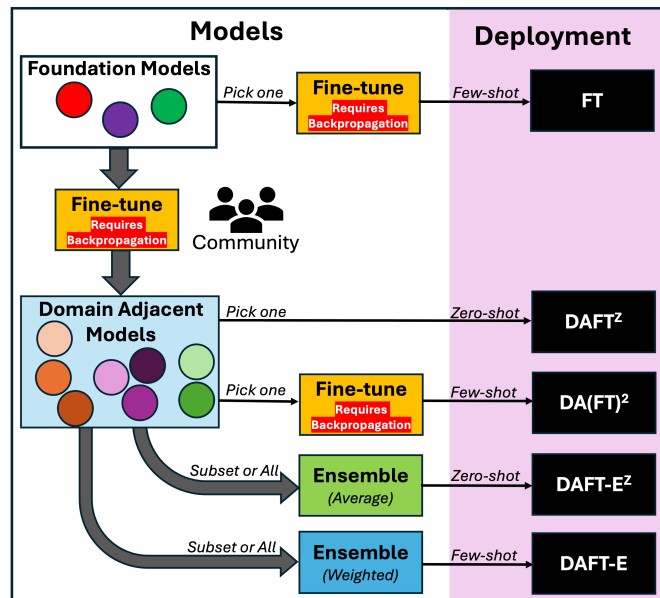

Figure 1: *Different options for few-shot tasks given LLMs and community created DAFT models*: (1) Directly fine-tune the LLM (FT); (2) Select one DAFT model and use it zero-shot (DAFT$^Z$); (3) Use a DAFT model but with task-specific fine-tuning (DA(FT)$^2$); (4) Ensemble multiple DAFT models and use it zero-shot (DAFT-E$^Z$); (5) Use ensemble of DAFT models with extremely lightweight few-shot adaptation of the ensemble weights (DAFT-E).

ated DAFT models. These four options along with fine-tuning the base model using data from target domain (FT in Fig. 1) are defined as follows:

1. **FT** – Fine-tune *base models* with training data obtained from $D_T$.
2. **DAFT$^Z$** – Pick one available DAFT model and use it for zero-shot inference (no training cost).
3. **DA(FT)$^2$** – Pick one available DAFT model and fine-tune with training data from $D_T$.
4. **DAFT-E$^Z$** – Use a subset or all DAFT models and ensemble them for zero-shot inference.
5. **DAFT-E** – Use a subset or all DAFT models for few-shot learning of ensemble weights using training data from $D_T$ and perform inference.

Section 2 discusses related work, followed by Section 3 discussing DAFT models and their potential for various NLP tasks. Section 4 discusses zero-shot and few-shot performance, while Section 5 theoretically characterizes conditions that result in strong performance from the DAFT ensemble.

## 2 Related Work

The two main aspects of our work are: (i) trying to adapt to data-scarce tasks or transfer learning, (ii) leveraging multiple pre-fine-tuned models via ensembling. Here we will briefly discuss related work on transfer learning, ensembling and the closely related blending and mixture-of-experts.

Transfer learning (Pan & Yang, 2009; Zhuang et al., 2020) is closely related to the goals of our work, as we wish to adapt a model trained on a data-rich source distribution to a data-scarce target problem. The success on the target task often relies heavily on the level of *positive transfer* one can achieve, and various ways of training the model on the source and target data have been developed to maximize the positive transfer (e.g., by extending the already expensive pre-training phase (Gururangan et al., 2020)). A thorough empirical study on the factors for good transfer learning across diverse domains and tasks has been done in Mensink et al. (2021). In Zamir et al. (2018), a parameterized readout function represented by a shallow network was introduced to check (after necessary training) the affinity between any two tasks. Given some budget constraints, using the affinity matrix, the performance on $T$ tasks can be optimized by only training models on $S$ tasks, where $T \gg S$. Here, we consider the DAFT models as the source models for knowledge transfer. However, not every DAFT model may transfer positively, since the DAFT models are obtained from public non-curated pools. Also, for few shot problems, fine-tuning DAFT with target domain data (denoted as $DA(FT)^2$) is a *transfer learning* option that is considered here ($DA(FT)^2$ in Fig. 1).

*Ensembling* is a technique that combines the predictions of multiple classifiers to generate a single decision and has been extensively investigated over the past few decades (Breiman, 1996; Clemen, 1989; Maclin & Opitz, 1997; Dietterich, 2000). The pre-requisites for effective ensembling are: (i) models with decent performance, and (ii) models that make independent errors, i.e., training diversity in models (Ovadia et al., 2019; Lakshminarayanan et al., 2017). The ensemble of models can be the weighted or unweighted average of the model outputs. For weighted averaging, some training is needed to learn the weights (Caruana et al., 2004). As another approach, Dvornik et al. (2020) proposed a method to select a weighted concatenation of the output features of models corresponding to multiple domains (models fine-tuned with different datasets) in a few shot environment. The optimized weights of the features are calculated by minimizing the negative log-likelihood classification loss on the few-shot target data. Isotropic merging (Polyak & Juditsky, 1992) is a technique similar to ensembling that has been studied alongside ensemble methods over the last few decades and has shown promising performance with LLMs, e.g., Model Soup and its variants (Wortsman et al., 2022). Recently, a variation of Isotropic merging has been introduced as AdaMerging (Yang et al., 2023), where multiple LLMs, individually expert on different tasks, are merged to generate a single model that can perform well on all tasks. A forward pass on some unlabeled data is needed to find the merging coefficients using entropy minimization. One of the main contributions of our work is to focus on understanding the efficacy of ensembling in the context of DAFT models for data-scarce tasks.

The main differences between Model Soup (Wortsman et al., 2022), AdaMerging (Yang et al., 2023), and the proposed ensemble methods (DAFT-E$^Z$ and DAFT-E), are highlighted in Table 1. In the table the following abbreviations are used; SLI: Single LLM inference, NMWA: No (LLM model) weight access, MAE: multi-architecture ensemble, SFPA: single forward pass (for few-shot) adaptation. The table highlights that DAFT-E$^Z$ and DAFT-E provide lightweight LLM backprop-free model combination schemes (similar to Uniform Soup, Greedy Soup and AdaMerging), without requiring access to the model weights, and having the ability to combine LLMs with different architectures. Furthermore, the few-shot adaptation in DAFT-E requires only a single forward-pass with the few-shot samples compared to Greedy Soup, which requires multiple forward-passes. However, both Uniform and Greedy Soup finally need to perform inference with a single LLM while DAFT-E$^Z$ and DAFT-E have to perform inference with multiple LLMs.

Another approach related to ensembling is *blending* of models. For instance, LLM-Blender (Jiang et al., 2023) performs ensemble of $n$ LLMs by pairwise ranking and generative fusion. The pairwise ranking approach requires creating a custom dataset, training a BERT model, and it incurs substantial computational overhead to perform inference. The proposed generative fusion combines the ranked list from the pairwise ranking with a fine-tuned LLM to generate a response. In contrast, our approaches with DAFT focus on a completely task agnostic (e.g., not just generative tasks) and computationally low-cost solution to leverage multiple fine-tuned LLMs (Please check Table 1 for comparison). Another type of blending is introduced in Lu et al.

(2024), and is orthogonal to ensemble. The method selects base models at random and combines them by adding the response of an already evaluated model to the input of the subsequent model; a sequential approach that can be readily combined with ensembling.

*Mixture of Experts* (MoE) (Jacobs et al., 1991) has been utilized by different ML models for both regression and classification tasks (Yuksel et al., 2012). Before recent developments in LLMs, MoE was mainly focused on dense mixture of experts, which has been replaced by a sparse mixture of experts (Fedus et al., 2022). One recent work on MoE, Mixtral (Jiang et al., 2024) uses a sparse LLM MoE where a routing network (e.g., Rosenbaum et al. (2017)) is pre-trained to map input tokens to a subset of experts. Compared to all these MoE models (which requires multiple backpropagation through all the involved LLMs to train the routing network), the ensemble of DAFT models that we consider require at most a single forward pass through the LLMs for few-shot target task adaptation, and hence are computationally significantly more lightweight.

Table 1: Comparison of different methods to combine pretrained models in terms of the features. **BPF**: Method is LLM backpropagation free. **SLI**: Method needs to perform inference on the target task with a single LLM. **NMWA**: No LLM model weight access is required by the method. **MAE**: Method can handle multi-architecture ensemble. **SFPA**: Method requires a single forward pass for few-shot adaptation.

| Method | 0-shot | BPF | SLI | NMWA | MAE | SFPA |
|---|---|---|---|---|---|---|
| DAFT | ✓ | ✓ | ✓ | ✓ | N/A | N/A |
| Uniform Soup | ✓ | ✓ | ✓ | ✗ | ✗ | N/A |
| AdaMerging | ✓ | ✓ | ✓ | ✗ | ✗ | N/A |
| DAFT-E$^Z$ | ✓ | ✓ | ✗ | ✓ | ✓ | N/A |
| FT | ✗ | ✗ | ✓ | ✗ | N/A | ✗ |
| DA(FT)$^2$ | ✗ | ✗ | ✓ | ✗ | N/A | ✗ |
| Greedy Soup | ✗ | ✓ | ✓ | ✗ | ✗ | ✗ |
| LLM-Blender | ✗ | ✗ | ✗ | ✓ | ✓ | ✗ |
| DAFT-E | ✗ | ✓ | ✗ | ✓ | ✓ | ✓ |

## 3 DAFT Models

With the rapid expansion of open platforms like Colab and Kaggle to perform model training, and having access to different public datasets, researchers can access LLMs and datasets that can be used to fine-tune these LLMs. This has lead to a large repository of publicly available *fine-tuned base models* (as defined in Section 1) (HF, 2024c; Kim, 2023). We formally define *base model* as;

**Definition 1.** *A **base model** has a pre-trained LLM and a task-specific (un-tuned) header layer added on top of the LLM.*

In Section 1 we argued that the performance of any such fine-tuned base model on some target dataset depends on the source dataset on which it was fine-tuned on. A base model's header layer can vary depending on the task, and it can become a domain adjacent fine-tuned model (DAFT) for a task when the base model is fine-tuned (either the header layer or the full model) with a dataset that has the same task criterion as the target data. So, we define the DAFT model as follows:

**Definition 2.** *A domain adjacent fine-tuned model (DAFT) is a fine-tuned base model, if the model can be used for the same (general) task as the target task.*

The task similarity between two tasks can be calculated in two ways: (i) by the source and target dataset similarity and (ii) by task description similarity. Since in most cases the source dataset, with which the DAFT model was fine-tuned, is not available, method (ii) is more general. For method (ii), the task similarity can be identified manually or measured using an LLM fine-tuned for 'textual similarity'. In our definition of DAFT, we are proposing to follow method (ii) to find DAFT models, i.e., when a model is fine-tuned with a dataset that has the same task criterion as the target data.

### 3.1 Formation of DAFT models

### 3.1.1 Tasks and Datasets

We consider two broad NLP tasks, (1) Sentiment Analysis (positive or negative sentiment) and 2) Textual Similarity (if two sentences are similar). Although there are (currently) 1667 and 936 DAFT models available for sentiment analysis and textual similarity task in Hugging Face platform, to mitigate the risk of benchmark data leakage, we chose to create our own DAFT models by fine-tuning the base models with specific

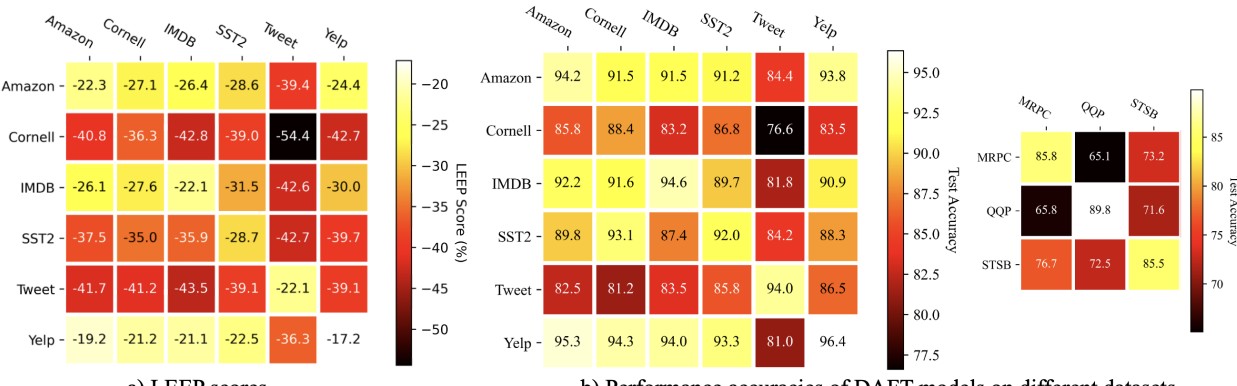

Figure 2: a) LEEP score heatmap: LEEP scores signifying the transferability of knowledge from one dataset to another for sentiment analysis datasets. b) *Efficacy of DAFT models*: The vertical axis corresponds to the data used to fine-tune a base LLM (here we show Roberta-base; the heatmaps for the other models show a similar trend and are provided in appendix), and the horizontal axis corresponds to the unseen test data. All datasets, for each of the two tasks, have matching output spaces respectively and pertain to *sentiment classification* and *textual similarity* problem respectively (left and right). The diagonal is the performance of FT– the LLM tested with data from the distribution it was fine-tuned with, denoting the upper watermark for that test data (only exception being SST2 – discussed in text). The off-diagonal entries correspond to the performance of the DAFT– LLM trained on one dataset, and tested on another, highlighting the potential computationally cheap (free!) zero-shot benefit that DAFT models can provide. Note that, for any given test data, the performance of a DAFT LLM can vary significantly. Also, LEEP score heatmap and DAFT models performance shows very similar trend.

datasets[1] (HF, 2024a; University, 2024; Kaggle, 2024). The datasets for Sentiment Analysis are: Amazon polarity, Cornell Movie, IMDB, SST2, Tweet sentiment, and Yelp Polarity; and for Textual Similarity: MRPC, QQP, STS-B (details in Appendix A.1).

### 3.1.2 Models

For both tasks, we chose three LLMs (with fine-tuning) for different performance analyses: (1) Roberta-base, (2) BERT-based-uncased, and (3) xlnet-base-cased. These models with a header layer form base models for our experiments.[2]

### 3.1.3 Base Models

A *base model* is created by adding a header layer to an LLM. Since the base models are not trained to perform any specific task and the header layer has not been fine-tuned, the performance of all the base models is similar to random guessing. For some base model $B$ and dataset $D$, let us denote $\Phi(B, D, n)$ as the fine-tuned version of $B$ on $n$ amount of data from dataset $D$. If the parameter $n$ is missing as the argument, the base model is assumed to be fully fine-tuned ($FFT$) on $D$. In our experiments, with few shot fine-tuning, we vary $n$ in the range of $2 - 256$ samples. For the ($FFT$) models, we fine-tune until loss stabilization[3].

### 3.2 Performance of DAFT Models

---

[1]The performance of DAFT models downloaded directly from Huggingface are analyzed further in Appendix A.8.

[2]These three models were chosen from a few other models due to their usually better performance (on the chosen datasets) compared to all the other models of the same size. We also used BART-LARGE-MNLI, ROBERTA-LARGE-MNLI, and OPT-1.3B models for sentiment analysis (zero-shot classification). These models are much larger in size compared to the above-mentioned models, and therefore needs more computational power for inference.

[3]Fine-tune until the fluctuation of loss is less than $\pm 1\%$.

Let us denote the train and test splits of the target dataset as $D'_T$ and $D''_T$ respectively. Since the standard train and test splits are identically distributed (iid), we expect a model trained on the target data ($D'_T$) to perform the best on the test data $D''_T$. This performance value is the *ceiling benchmark* that we aim to match or surpass. For that purpose, we performed full fine-tuning ($FFT$) on all three base models using training data of all datasets (sentiment analysis and textual similarity), to come up with 18 $FFT$ models for sentiment analysis and 9 $FFT$ models for textual similarity. Hence, for any target dataset, we can have 15 DAFT models for sentiment analysis by excluding the three models that were fully fine-tuned on $D'_T$, and 6 DAFT models for textual similarity task.

We plot the zero-shot performance of model fine-tuned on one dataset and tested on another for both sentiment analysis and textual similarity tasks in Fig. 2 (b). It is important to note from the figure that most of the zero-shot performances are significantly better ($\gg 50\%$) than the performance of a model with untrained header layer (50% for binary

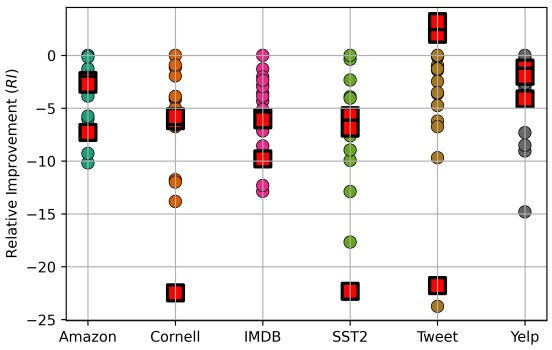

Figure 3: *RI of DAFTs compared to the Single Best DAFT* for *sentiment analysis*: For each test dataset, we consider 15 DAFT LLMs (fine-tuned on data different from the test data), and show the RI of DAFT$^Z$ over the single-best DAFT LLM (out of the 15) (the colored circles ●). Values less than 0 indicate performance degradation. The red squares ■ denote the performance of zero-shot classification with larger LLMs, i.e., BART-LARGE-MNLI, ROBERTA-LARGE-MNLI, and prompting with OPT-1.3B.

classification). We also note that the performance of the Cornell and Tweet datasets do not match the other four datasets' performance on the sentiment analysis task. Our empirical study suggests that the test data of Cornell is distributionally different from the other datasets, and for all FFT models the performance was poor. The Tweet dataset seems to have a different distribution in both its training and testing data (compared to the other 5 datasets). From the Tweet row and column, we observe that apart from the diagonal element, all the other accuracies are small, and a high 94% accuracy is attained when the model is trained and tested on Tweet data. Upon inspecting the row of SST2, another interesting observation can be made, the performance of DAFT-Cornell beats that of DAFT-SST2. This is caused by the test dataset of SST2 being more aligned with the training data of Cornell (further details in Appendix A.3). For textual similarity datasets, the performance of DAFT models was not as good as the sentiment analysis, but still show promise. Lastly, to validate the transferability of the DAFT models to a new target domain or task, we looked at the LEEP scores (Nguyen et al., 2020) for all the 6 datasets of the sentiment analysis task (Fig. 2 (a)). Interestingly, the heat map with the LEEP scores closely follows the heat map that we generated using the performance of the DAFT models (Fig. 2 (b)). Thus, the transferability of DAFT models (corresponding LEEP scores) to a target domain is proportional to their (inference) performance on that target domain. These LEEP scores also tells us that a LEEP scored based ensemble method is possible and needs further investigation.

To demonstrate practical feasibility of DAFT models, we explored *readily available* DAFT models from public repositories. For target task we chose it to be question/answer in the field of *Medical Knowledge* and selected 9 sub-datasets from the MMLU dataset to be domain-adjacent of the target task (judging from the dataset names). These 9 sub-datasets are: Clinical Knowledge (CLNC KNW), College Biology (CLG BIO), Professional Medicine (PRF MED), Virology (VIROL), Nutrition (NUTR), Anatomy (ANATOM), College Medicine (CLG MED), High School Biology (HGH SCH BIO), and Medical Genetics (MED GEN). For each of these sub-datasets, we searched for corresponding fine-tuned models in public repositories. We found that *LoRA*-tuned DAFT models were available for all of them, with Llama 3.1 serving as the base model. After evaluating the performance of these DAFT models on all 9 sub-datasets, we created a heatmap following a similar method used for Fig. 2(b), and the result is shown in Fig. 4. In the heatmap, *columns* represent the publicly available (LoRA-tuned) DAFT models, and *rows* correspond to the individual sub-datasets. It is noticeable that these DAFT models exhibit highly variable performance across different sub-datasets; however, for any specific sub-dataset, the performance across different DAFT models does not vary significantly.

Table 2: *Average Relative Improvement of DA(FT)$^2$ compared to FT* (Top 6: sentiment analysis tasks. Bottom 3: textual similarity tasks): Positive values show DA(FT)$^2$ improvement over FT on target (domain); larger values imply higher improvements. Please see Table 9 and 10 in Appendix for detailed results. We see that DA(FT)$^2$ is significantly better than FT in few-shot problems. (The same *RI* values are plotted as line plot on the right to capture the trend with training data size.)

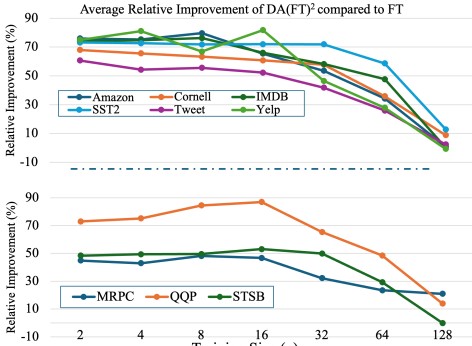

| Target | 2-shot | 4-shot | 8-shot | 16-shot | 32-shot | 64-shot | 128-shot |
|--------|--------|--------|--------|---------|---------|---------|----------|
| Amazon | 75.95 | 75.25 | 79.62 | 65.40 | 53.51 | 34.26 | 0.81 |
| Cornell | 67.95 | 65.61 | 63.26 | 60.72 | 57.77 | 35.85 | 8.85 |
| IMDB | 74.26 | 74.64 | 76.19 | 66.01 | 58.20 | 47.64 | 0.10 |
| SST2 | 73.13 | 72.64 | 71.81 | 71.95 | 71.88 | 58.58 | 12.79 |
| Tweet | 60.67 | 54.25 | 55.53 | 52.25 | 41.82 | 25.92 | 2.34 |
| Yelp | 74.88 | 81.00 | 66.81 | 81.68 | 46.47 | 27.86 | -0.72 |
| MRPC | 44.88 | 42.92 | 48.08 | 46.74 | 32.15 | 23.47 | 20.90 |
| QQP | 72.93 | 75.17 | 84.44 | 86.97 | 65.36 | 48.50 | 13.86 |
| STSB | 48.36 | 49.31 | 49.52 | 53.08 | 49.88 | 29.24 | -0.12 |

DAFT models *can potentially* provide competitive zero-shot performance on new tasks.

### 3.2.1 Performance of DAFT$^Z$ *(Zero Shot)*

Fig. 3 shows the relative improvement (*RI*) of all the DAFT models for sentiment analysis task when compared to the single best performing DAFT model.[4] From Fig. 3 it is evident that there are some DAFT models that are not as good as the single best and for some specific choices (as for Tweet), the performance can be poor. On the other hand, for zero shot classification with BART-LARGE-MNLI, ROBERTA-LARGE-MNLI, the performance were quite well. However, the zero-shot prompting using OPT-1.3B showed poor performance on most of the datasets. However, these (8 times) larger models (with zero shot classification or prompting) still fail to beat the single best DAFT in all cases, except Tweet. The low performance of DAFT models on Tweet is potentially because of the difference in the nature of this dataset (discussed previously in Section 3.2 and is supported by Fig. 2).

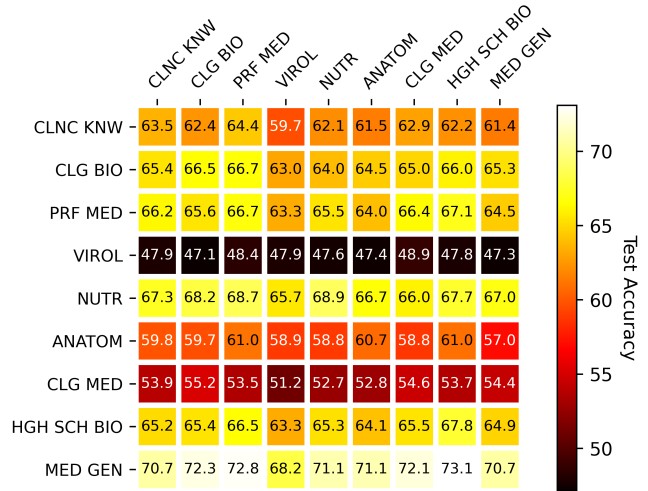

Figure 4: Performance of DAFTs on MMLU medical tasks: The 9 datasets are related to medical tasks, and the corresponding 9 DAFT models were obtained from Hugging Face.

Zero-shot performance of DAFT models can vary significantly, with some DAFT models even outperforming larger LLMs, making the choice of the DAFT model critical.

### 3.2.2 *Few Shot* Comparison of FT & DA(FT)$^2$

When (some) training data is available from the target domain, the general approach is to fine-tune a model with that data to perform the domain specific task. The key research question is: should we fine-tune a base model (FT in Fig. 1), or fine-tune a DAFT (DA(FT)$^2$ in Fig. 1)? We explored two scenarios, with weight updates of i) only the header layer connection and ii) the whole model, and decided to go with the weight

---

[4]Thus, all the *RI* values (corresponding to the other DAFTs) will be non-positive (zero for the best single DAFT). The relative improvement (*RI*) of all the DAFT models for textual similarity task is provided in Appendix (Fig. 19).

update of the whole model [5]. Table 2 provides the average performance comparison of the FT and DA(FT)$^2$ models using the *Relative Improvement* (*RI*) metric when the number of samples to fine-tune is varied from $n = 2$ to 128 (Full results are in Appendix, Table 9 and 10) [6]. A positive (negative) value of *RI* means DA(FT)$^2$ is performing better (worse) compared to FT, and the larger (smaller) the value the better (worse) DA(FT)$^2$ is performing. We observe that DA(FT)$^2$ outperforms FT by a large margin when $n$ is small (2 to 64), and see only one negative value in the table ($n = 128$). This indicates that DA(FT)$^2$ is outperforming FT in most (few shot) cases on the target task. Please note that if there are a moderate amount of data-samples (for the current tasks, $n > 128$) available from the target domain (not a data-scarce environment) FT is expected to perform similarly as DA(FT)$^2$.

> For few-shot tasks, fine-tuning DAFT models, that is DA(FT)$^2$, is generally more beneficial than fine-tuning base models (FT).

From a practical standpoint, a major advantage of using DAFT models is that we get them for *free.* However, as our results show, there is considerable uncertainty about what is the right DAFT model to choose (especially in the zero-shot setting), as the performance can significantly vary between different DAFT models (Fig. 3). Also, note that while DAFT$^Z$ is zero shot, DA(FT)$^2$ is computationally expensive, needing back-propagation. Hence, in the next section, we explore the use of ensemble methods, which is a cheaper alternative and addresses the model selection problem.

## 4   Ensembling of DAFT Models

In this section, we evaluate the performance of the DAFT models using two distinct *Ensemble* methods and benchmark their performances. Advantages of using ensemble methods are discussed in Appendix A.5.

### 4.1   DAFT-E$^Z$ Performance *(Zero Shot)*

The average Ensemble (DAFT-E$^Z$) method is a zero-shot method that can be used as an alternative to randomly picking one of the DAFT models. In DAFT-

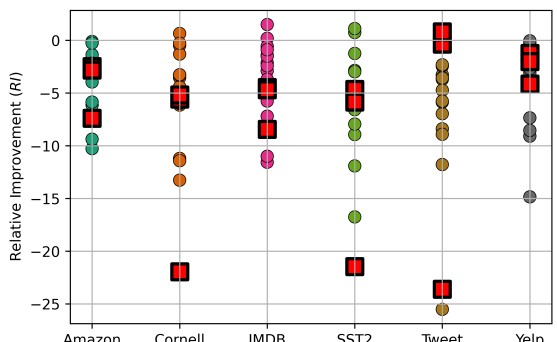

Figure 5: *RI of DAFTs over DAFT-E$^Z$ for sentiment analysis*: For each test dataset, we consider 15 DAFT LLMs (fine-tuned on data different from the test data), and consider the RI of DAFT$^Z$ over DAFT-E$^Z$ of 15 DAFT LLMs. Values less than 0 indicate performance degradation. The red squares ■ denote the performance of zero-shot classification on larger LLMs (BART-LARGE-MNLI, ROBERTA-LARGE-MNLI), and prompting with OPT-1.3B. The results show that DAFT-E$^Z$ usually has significantly better zero-shot performance than DAFT$^Z$. However, there are some cases where some specific DAFT models can outperform the average ensemble (leading to RI > 0).

E$^Z$, the *output probabilities* from all the DAFT models are averaged and the decision is taken on that value (max-voting is another feasible approach). To perform DAFT-E$^Z$, we run inference on all the DAFT models. However, since these DAFT models do not need any further training or fine-tuning and inference can be run in parallel, it is feasible to utilize them efficiently with the abundance of computational devices (mostly low end) available today.

Fig. 5 shows the *Relative Improvement* of all the 15 DAFT models and the three larger models (as previously discussed in 3.1.2) for sentiment analysis, compared to the average ensemble of the DAFT models, i.e., DAFT-E$^Z$. There are very few cases (5/90 for DAFT$^Z$, and 1/18 for larger LLMs) in Fig. 5 where a positive *RI* value is seen, implying that the average ensemble method (DAFT-E$^Z$) is better than most of the DAFT models. Moreover, the performance of the larger LLMs (red squares) are not as good as DAFT-E$^Z$. Furthermore,

---

[5]We found that updating the weights of the header layer only does not guarantee good performance and can be surpassed by models designed to perform similar tasks without any fine-tuning (zero-shot). So, we chose to do all performance analysis with fully fine-tuned models. Performance comparison of updating the weights of the entire model versus only the header layer is provided in the Appendix (Table 9).

[6]The *RI* values in the table are generated with *Roberta* as the base models for both DA(FT)$^2$ and FT.

Table 3: Average $RI$ of $DA(FT)^2$ compared to DAFT-E (Top 6: sentiment analysis tasks. Bottom 3: text similarity tasks). Negative values indicate that DAFT-E is better than $DA(FT)^2$ on target (task); larger values imply higher improvements. Table 11 and 12 in the Appendix gives more details. In the very few-shot setting, DAFT-E outperforms $DA(FT)^2$ in almost all cases.

| Target | 2-shot | 4-shot | 8-shot | 16-shot | 32-shot | 64-shot | 128-shot |
|---|---|---|---|---|---|---|---|
| Amazon | -4.03 | -2.48 | -2.38 | -2.42 | -2.66 | -2.95 | -2.06 |
| Cornell | -3.94 | -3.60 | -3.37 | -3.54 | -4.13 | -3.91 | -2.53 |
| IMDB | -2.70 | -3.41 | -3.35 | -3.31 | -3.04 | -2.87 | -2.83 |
| SST2 | -4.67 | -3.32 | -3.94 | -4.87 | -4.11 | -4.68 | -4.21 |
| Tweet | -4.62 | -4.91 | -5.14 | -5.54 | -4.00 | -3.82 | -1.59 |
| Yelp | -2.56 | -2.76 | -2.10 | -2.09 | -1.68 | -2.46 | -2.28 |
| MRPC | -5.19 | -4.34 | -4.11 | -3.60 | -3.12 | 1.53 | 4.74 |
| QQP | -2.85 | -2.64 | -0.46 | -0.06 | 2.01 | 2.41 | 4.42 |
| STSB | -5.29 | -5.77 | -4.07 | -1.89 | -1.97 | 0.57 | 1.28 |

for the Tweet dataset, for which none of the DAFT models performed well and was beaten in performance by two of the larger models (*Bart-large-mnli*, *Roberta-large-mnli*), we now have $RI$ values close to zero. Furthermore, for the zero-shot Tweet task, while the larger models outperformed all the $DAFT^Z$ models (Fig 5), $DAFT-E^Z$ is able to match their performance, highlighting the utility of ensembles. Hence, $DAFT-E^Z$ has a performance very close to those larger models and shows the importance of using an ensemble even when individual DAFT models cannot perform well. For text similarity task we see a similar trend in the $RI$ performance (Fig. 6).

Lastly, to evaluate the performance of $DAFT-E^Z$ on practical and modern LLM benchmark datasets, we performed an ensemble of 8 DAFT models across the 9 datasets selected for the target task of *Medical Knowledge* (see Section 3.2 for details on the datasets). The results are shown is Fig. 7. We observe that $DAFT-E^Z$ outperforms the base model (Llama 3.1) on all instances (red squares with black border) and surpasses individual DAFT models in 56 out of 72 cases (circles in the figure). In Section 5, we argue theoretically (Proposition 1) that it is better (in an expected sense) to use $DAFT-E^Z$ over choosing a DAFT model randomly from the set of DAFTs.

> For zero-shot tasks, just a simple ensemble of DAFT models ($DAFT-E^Z$) is on average a better choice than any individual DAFT model.

The average ensemble is therefore a great data-agnostic solution; however, when training data from the target domain is available, we show that the data-informed ensembling can be a stronger choice. We thus pursue a weighted ensemble of the DAFT models (DAFT-E) in the following section.

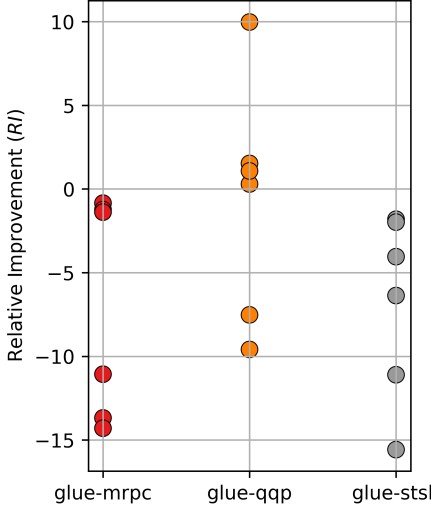

Figure 6: *Relative Improvement of DAFTs compared to DAFT-E$^Z$ for Text similarity task:* For each test dataset, we consider 6 DAFT FMs (fine-tuned on data different from the test data), and consider the RI of $DAFT^Z$ over the $DAFT-E^Z$. Values less than 0 indicate better performance of $DAFT-E^Z$.

## 4.2 *Few-Shot* DAFT-E and $DA(FT)^2$

We define DAFT-E as the weighted ensemble method, where the weights of the ensemble layer are learned using the training data from the target dataset. To perform the weighted ensemble in DAFT-E we utilized two specific regression methods, a) Random forest-based regression ($RF$) and b) SGD-based linear regression ($LR$)[7]. Since, $LR$ is more lightweight with comparable or better performance than $RF$, we considered $LR$

---

[7]Parameters of these methods are given in Appendix A.6.

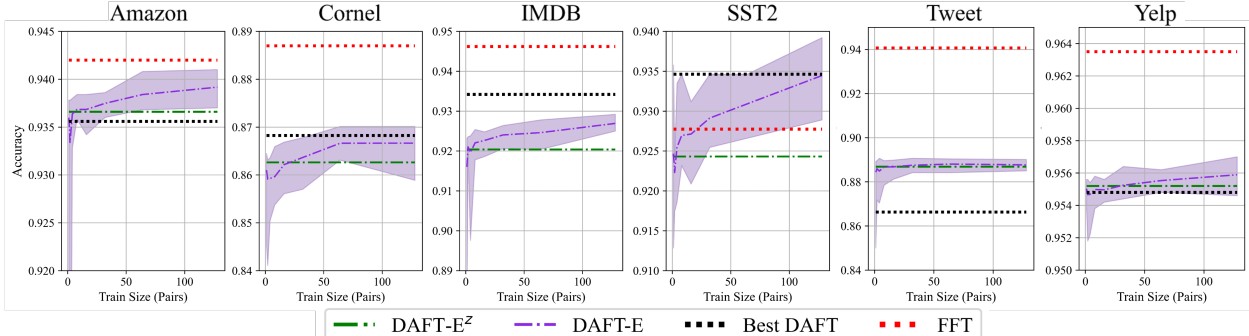

Figure 8: *Performance comparison of DAFT-E$^Z$, DAFT-E, single-best DAFT and FFT.* Error interval for DAFT-E is based on the random choice of the few-shot samples used to learn the weights of the ensemble aggregated over 10 trials. The single best DAFT for different datasets: Amazon (Roberta - Yelp), Cornell (Roberta - SST2), IMDB (xlnet - amazon), SST2 (xlnet - Cornell), Tweet (xlnet - SST2), Yelp (xlnet - amazon).

to generate the results of DAFT-E (More discussion on $LR$ and $RF$ performance comparison with the results are in Appendix: Table 11 and 12).

In Section 3.2.2, we observed that DA(FT)$^2$ usually performs better than FT for few-shot learning. Hence, we compare the performance of DAFT-E (weighted ensemble) with DA(FT)$^2$. For a fair comparison, the same amount of data is used when fine-tuning DA(FT)$^2$ and learning the weights of the ensemble layer. Table 3 shows the average RI of DA(FT)$^2$ compared to DAFT-E. An *RI* of positive (negative) means that DA(FT)$^2$ performed better (worse, resp.) compared to DAFT-E. From Table 3 it is evident that DAFT-E outperforms DA(FT)$^2$ for sentiment analysis tasks, i.e., *all* values are negative in the table. On the other hand, for textual similarity DAFT-E usually performs well for smaller data samples but underperforms on more-data (positive *RI* values). However, $RF$ performed well with more samples and actually performs better or similarly to DA(FT)$^2$ for $n > 16$ (Table 13 in the Appendix). This shows that weighted ensembling is effective, with task-specific variations between $RF$ and $LR$. Note that updating the ensemble layer depends on the predictions of the DAFT models, and requires solv-

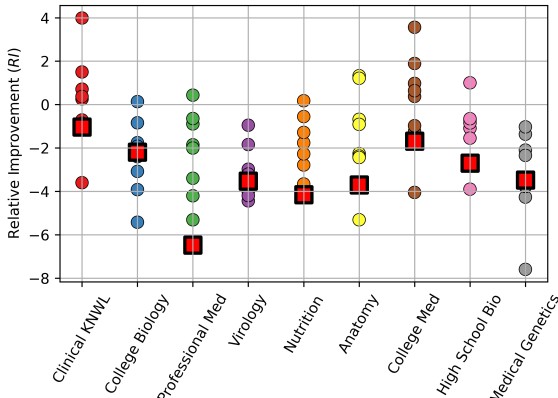

Figure 7: *Relative Improvement of DAFTs compared to DAFT-E$^Z$* for *Medical Knowledge* task: For each dataset, we consider 8 DAFTs, and consider the RI of DAFT$^Z$ over the DAFT-E$^Z$. Values less than 0 indicate better performance of DAFT-E$^Z$.

ing either (i) a simple linear equation to optimize the weights of the ensemble layer, or (ii) training a $RF$ with shallow trees. Thus, *only a single inference on each of the DAFT models* with the few-shot samples is necessary [8]. In DA(FT)$^2$, where we fine-tune a DAFT model, we need to perform *computationally expensive* back-propagation through the large model to update weights.

> For few shot tasks, (cheap) *weighted* ensemble of DAFTs (DAFT-E) can outperform (expensive) DA(FT)$^2$.

### 4.3 Overall Comparison

---

[8]In our evaluations, we found that DAFT-E puts much more weight on the top few DAFT models. Hence if we have some training data from the target domain and have some budget constraints (e.g., the number of models), one solution could be to pick the models with the highest weights (from DAFT-E LR). Moreover, other transferability metrics can be used to select the top models within a budget. Some of those transferability metrics are LEEP score, MS-LEEP, E-LEEP, and SoftIoU-EEP (Agostinelli et al., 2022). LEEP scores can be calculated for each model individually (as discussed in Section 3). The other metrics, i.e., MS-LEEP, E-LEEP, and SoftIoU-EEP, are computed for a set of models and can be an alternative method to LR in DAFT-E to ensure better transferability for an ensemble of models.

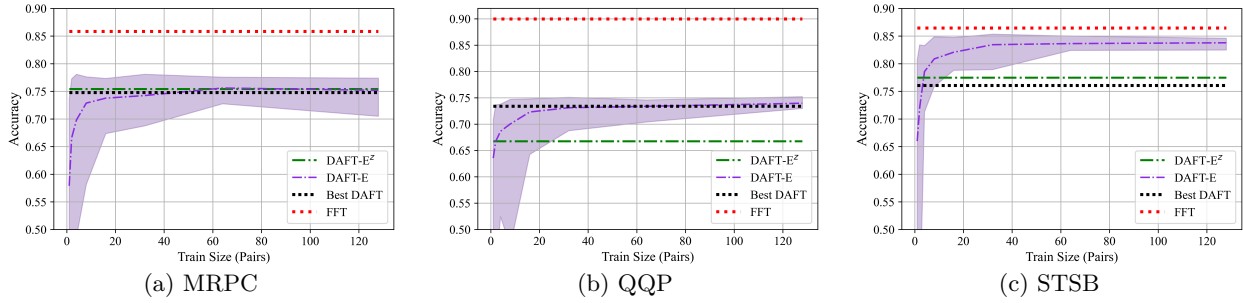

(a) MRPC         (b) QQP         (c) STSB

Figure 9: *Performance comparison of DAFT-E$^Z$, DAFT-E, single-best DAFT and FFT* for Text Similarity task. The error interval for DAFT-E is based on the random choice of the few-shot samples used to learn the weights of the ensemble, using RF method, aggregated over 10 trials. The single best DAFT for different datasets are: MRPC (Roberta - STSB), QQP (Roberta - STSB), STSB (Roberta - MRPC).

For benchmarking purposes, let us assume that we have the entire training data ($D_T$) of the target domain, and have a model that is fine-tuned on $D_T$ (*FFT* on $D_T$). Let us compare the performance of the ensemble methods with this FFT on different datasets. Fig. 8 and 9 show the performance comparison of DAFT-E$^Z$ (green dash-dot line), DAFT-E (purple dash-dot line), single best DAFT (black dotted line), and FFT on $D_T$ (red dotted line) for all six datasets of the sentiment analysis and for all three datasets of the text similarity tasks respectively. From Fig. 8 and 9, we observe the advantage of using DAFT-E over DAFT-E$^Z$ when training data is available. It is interesting to note that DAFT-E has

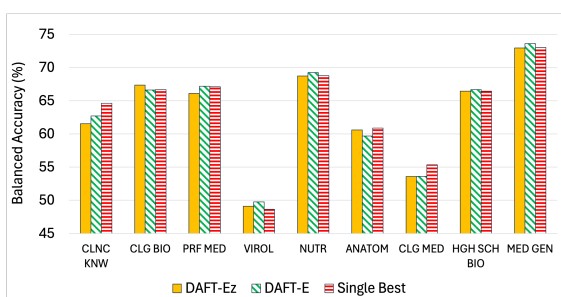

Figure 10: Performance of DAFT-E$^Z$, DAFT-E and Single Best DAFT for Medical Knowledge task with MMLU datasets.

a strong increasing trend in performance (w.r.t. fine-tuning data size) for most cases, and catches up with either the single best DAFT or FFT for the target domain with the increase of fine-tuning data. The performance of DAFT-E compared to the best solution, i.e., FFT on the target domain, can be theoretically bound and is given by Proposition 2 in Section 5.

Table 4 shows the overall comparison of the computational cost of all the five options that we have discussed in this paper. It should be noted that none of the ensemble methods suffer from the high computational complexity of back-propagation computation; further, using sparse weighting we can reduce the inference cost in DAFT-E compared to DAFT-E$^Z$. It is straightforward to see that DAFT$^Z$ and DAFT-E$^Z$ are the only ones among the five options discussed here that are zero-shot, and hence incurs zero training cost. In contrast, FT and DA(FT)$^2$ has fine-tuning cost, i.e., back-propagation cost ($C_B$), along with forward propagation cost ($C_F$). For DAFT-E, the training cost is the forward propagation cost of $N$ DAFT models along with the linear cost for learning the weight of the en-

Table 4: Computational cost of methods shown in Fig. 1. $n$ denotes the number of few-shot samples. $C_F$ and $C_B$ denote the computational costs of a forward and backward pass for a LLM. $E$ is the number of fine-tuning epochs. $N$ is the number of DAFT models available to ensemble, $\bar{N} \leq N$ is the number of nonzero weights in the weighted ensemble.

| Method | Zero-shot | Training cost | Inference cost |
|---|---|---|---|
| FT | ✗ | $n(C_F + C_B)E$ | $C_F$ |
| DAFT$^Z$ | ✓ | $0$ | $C_F$ |
| DAFT-E$^Z$ | ✓ | $0$ | $N \cdot C_F$ |
| DA(FT)$^2$ | ✗ | $n(C_F + C_B)E$ | $C_F$ |
| DAFT-E | ✗ | $Nn(C_F + E)$ | $\bar{N} \cdot C_F$ |

semble layer. Lastly, FT, DAFT$^Z$ and DA(FT)$^2$ use only a single (DAFT) model, and hence incurs only a forward propagation cost ($C_F$), whereas DAFT-E$^Z$ and DAFT-E use $N$ and $\tilde{N}$ DAFT models respectively in the inference stage.

To conclude the overall comparison, we present the performance of DAFT-E$^Z$, DAFT-E, and the single best DAFT model for the Medical Knowledge task using the MMLU datasets in Fig. 10. For DAFT-E, each dataset was split in half: one half was used to tune the linear weights of DAFT-E, and the other half was used for

Table 5: Performance Comparison of DAFT-E$^Z$ and Uniform Soup. The results show that, for zero-shot sentiment analysis tasks, DAFT-E$^Z$ performs at par or better than Uniform Soup. Note that Uniform Soup requires all models to have matching architectures so as to be able to uniformly combine the model weights. DAFT-E$^Z$ (and DAFT-E) do not have any such requirement.

| Dataset | Uniform Soup | DAFT-E$^Z$ |
|---------|--------------|------------|
| Amazon  | 93.29%       | **93.72**% |
| Cornell | 85.09%       | **85.37**% |
| IMDB    | 92.09%       | **92.28**% |
| SST2    | 88.07%       | **89.55**% |
| Tweet   | 86.09%       | **87.33**% |
| Yelp    | 94.67%       | **95.30**% |

Table 6: Robustness Comparison of DAFT-E$^Z$ and Uniform Soup in terms of the set of models being combined. The results show that as we increase the number of DAFT models, the performance of both Uniform Soup and DAFT-E$^Z$ improves, with DAFT-E$^Z$ consistently outperforming Uniform Soup in the zero-shot setting. As discussed in Table 5, we are able to add many more models into DAFT-E$^Z$ than in Uniform Soup because Uniform Soup requires all models in the ensemble to have matching architectures.

| DAFT models combined | Uniform Soup | DAFT-E$^Z$ |
|----------------------|--------------|------------|
| SST2, Tweet | 88.78% | **89.09**% |
| SST2, Tweet, Yelp | 90.57% | **90.86**% |
| SST2, Tweet, Yelp, Cornell | 91.19% | **91.39**% |
| SST2, Tweet, Yelp, Cornell, Amazon | 92.09% | **92.28**% |

performance evaluation[9]. Since DAFT-E$^Z$ and DAFT models required no tuning, only the second half of the data was used for evaluation. This ensured that all three methods were evaluated on the same data. Due to the small size of each dataset, we repeated the random split 200 times for each of the 9 datasets and report the average performance. From the figure, we observe that DAFT-E outperformed DAFT-E$^Z$ in 7 out of the 9 tasks. Most notably, DAFT-E even outperformed the single best DAFT model in 5 of the 9 cases, demonstrating the effectiveness of the DAFT-E approach.

> DAFT-E$^Z$ and DAFT-E are strong lightweight methods for leveraging already available DAFT models, often matching the single best DAFT model (only known post-hoc), and at times match the best possible performance of full fine-tuning with a large fine-tuning set.

### 4.4 Comparison with Model Soup

We compare the performance of DAFT-E$^Z$ and DAFT-E to a state-of-the-art zero shot ensemble model, Uniform Soup (Model Soup) (Wortsman et al., 2022) [10]. To do performance comparison between the zero-shot version of model soup, i.e., Uniform soup with DAFT-E$^Z$, we used the sentiment analysis task. We observed that when there are 5 DAFT models from some specific architecture (we could not use all 15, since there were 3 different architectures)[11], DAFT-E$^Z$ performs better than Model Soup (Table 5). To check the robustness of the DAFT-E$^Z$ method, we then changed our experiments to check performance of DAFT-E$^Z$ and Uniform soup on IMDB dataset by adding DAFT models to the respective methods. The result is shown in Table 6. Interestingly, DAFT-E$^Z$ showed more resiliency than Uniform soup.

## 5 Theoretical Analysis

**Notations.** The dataset of the target domain is $D_T$, the input data to the model is $\mathbf{x}$ and the output data is $\mathbf{y}$, which the model should predict given the input data. There are $N$ number of DAFT models, and a DAFT model can be built using any base model $B_j \in \mathcal{B}$, with $j = \{1, 2, ..., J\}$ and fine-tuning it on dataset $D_k \in \mathcal{DA}$ with $k = \{1, 2, ..., K\}$, where $\mathcal{DA}$ is the set of domain adjacent datasets. The $i^{th}$ DAFT model is given by $M_i$ and the corresponding base model and fine-tuning dataset are given by $B_{\kappa(i)}$ and $D_{\nu(i)}$, respectively, i.e., $B_{\kappa(i)}$ was fine-tuned on $D_{\nu(i)}$ dataset to get $M_i$. Here, $\kappa(i)$ and $\nu(i)$ are index

---

[9]DAFT-E performance reported here uses the same set of hyper-parameters for ensemble-weight learning across all datasets; dataset-specific tuning could further improve results.

[10]Methods like mixture of experts (MoE) need training with data and our motivation for ensemble was to avoid any training that needs expensive back-propagation. Hence, we refrained from comparing with techniques that require back-propagation unless it is fine-tuning with in-domain data, i.e., FT and DA(FT)$^2$.

[11]Model Soup only works if all the (DAFT) models are of the same architecture, which is not the case for DAFT-E$^Z$ (or DAFT-E). Furthermore, Model Soup needs all the weights of the models to work, but for ensembling, we do not need that. Just API access to the DAFT models is adequate.

mapping functions. Also, if we assume that all these `DAFT` models were created by fine-tuning fully on the given datasets (no fractional fine-tuning), then the total number of `DAFT` can not be greater than $J \times K$, or $N \leq JK$. Lastly, let $\ell(M(\mathbf{x}), \mathbf{y})$ be the loss in output when model $M$ is used on target dataset input $\mathbf{x}$ and the output is compared with $\mathbf{y}$. Since $\mathbf{x}$ and $\mathbf{y}$ are common for all loss calculation, we can just use $\ell(M)$ to represent $\ell(M(\mathbf{x}), \mathbf{y})$.

## 5.1 DAFT-E$^Z$ versus DAFT$^Z$

Determining if a `DAFT` model is going to perform well or not is very hard to know a priori if adequate time or data to train or test the model is not available. Hence, instead of choosing one of the `DAFT` models in random (`DAFT`$^Z$), one solution could be to use the ensemble of the output results (`DAFT-E`$^Z$) from all the `DAFT` models.

**Proposition 1.** *The performance of the average ensemble of DAFT models (DAFT-E$^Z$) is no worse than the expected performance obtained from choosing the DAFT models uniformly at random.*

Proposition 1 states that in terms of expected performance, using `DAFT-E`$^Z$ is no worse (strictly better, if $\ell$ is assumed to be strictly convex) than picking from the `DAFT` models uniformly at random. From the performance of `DAFT-E`$^Z$ shown in Fig. 5 and 6, we see that Proposition 1 holds, i.e., `DAFT-E`$^Z$ has better performance than the average performance of the `DAFT` models. Moreover, for these two specific types of tasks, the ensemble of the `DAFT` models beats the performance of individual `DAFT` models in most cases (98 out of 108 in total for both tasks).

## 5.2 DAFT-E versus Optimum

Let us denote $\tilde{M}_i$ as the model that uses the base model $B_{\kappa(i)}$ and is fine-tuned on $D_T$, i.e., $\tilde{M}_i = \Phi(B_{\kappa(i)}, D_T)$. Also, we denote $\tilde{M}_*$ is the best performing model when fine-tuned on $D_T$ (the base model for $\tilde{M}_*$ can be from $\mathcal{B}$ that are used to generate `DAFT` models or some other large model). The following proposition bounds the difference of loss between the optimum solution and `DAFT-E`.

**Proposition 2.** *The loss of DAFT-E is bounded as:*

$$\ell(\textit{DAFT-E}) \leq \ell(\tilde{M}_*) + \min_{i \in \mathcal{N}} \left[ \mu(\tilde{M}_i, \tilde{M}_*) + \rho(D_{\nu(i)}, D_T) \right], \tag{1}$$

*where $\mu(\tilde{M}_i, \tilde{M}_*)$ is the performance difference between the models $\tilde{M}_i$ and $\tilde{M}_*$ both fine-tuned on $D_T$, and $\rho(D_{\nu(i)}, D_T)$[12] is an appropriately defined distance measure between $D_{\nu(i)}$ and $D_T$.*

Usually base models (derived from FMs which are all quite large) perform similarly when fine-tuned on the full target dataset. Under that assumption, we can ignore the $\mu()$ term in the bound, resulting in the following corollary.

**Corollary 1.** *The loss of DAFT-E is larger than the optimum loss by no greater than $min_i \rho(D_{\nu(i)}, D_T)$, with the assumption that the base models of the DAFTs can perform as well as the optimum when fine-tuned on the target domain.*

Proposition 2 and Corollary 1 imply that the performance of `DAFT-E` depends on the base model on which the `DAFT` models were trained and the datasets they were fine-tuned on. If the base models (FMs) in the ensemble are nearly as good as the best possible base model (FM) (when the performance of the *FFT* of the base models are compared), then we can ignore the $\mu()$ term in Equation 1, and the performance of `DAFT-E` depends *only on the dataset that is closest* to the target domain. If the `DAFT` models are generated from a large number of adjacent datasets from diverse domains, we expect the distance from the target dataset to the closest `DAFT` dataset to be small, resulting in `DAFT-E` performing very close to the optimum.

In our experimental setting, the assumption made for the Corollary 1 holds, that is, all three base models are comparable in terms of performance when fine-tuned on the same dataset. Hence, the performance of

---

[12]Some practical dataset distance measurement metric are the Wasserstein distance metric (Panaretos & Zemel, 2019), Jensen-Shannon Divergence (JSD), etc.

`DAFT-E` depends on the adjacency of the datasets used in the `DAFT` models with the target dataset. For the case of sentiment analysis, we have six datasets, whereas for the text similarity task, we only have three. Thus, it is more probable that the `DAFT` models of the text similarity task are more diverse and `DAFT-E` will be able to achieve close to optimum performance. From the results (Fig. 8, and 9) it is evident that `DAFT-E` was able to perform much better for the sentiment analysis task, compared to the textual similarity task.

## 6 Conclusion

In this paper, we explore 4 ways to leverage the abundance of publicly available fine-tuned LLMs for data-scarce tasks. We investigate how the `DAFT` models can be utilized for inference under limitations on computation time or data required for training a model on a target task. The `DAFT` models can be used on the target domain (zero-shot) without any fine-tuning. If chosen properly, the performance of `DAFT` models on the target domain can be close to the optimal performance. However, this performance depends strongly on the choice of the `DAFT` model, and ensemble methods can be used to address this issue. Two ensemble methods, `DAFT-E`$^Z$ and `DAFT-E` are explored; their performances are empirically evaluated and compared against individual `DAFT` models and other benchmarks involving larger LLMs and base models trained with the full target domain dataset. Our theoretical results support the conclusions from the empirical findings, and provide insights under what conditions `DAFT-E` can provide near-optimal performance.

### Broader Impact Statement

The use of `DAFT` models can be a great low-resource and easy-to-use solution in data-scarce domains. Due to the current surge of open source `DAFT` models for different downstream tasks, the community can use these models directly without any (computationally expensive) fine-tuning of LLMs. The main innovation of the proposed ensemble methods, i.e., `DAFT-E`$^Z$ and `DAFT-E`, is the utilization of readily available `DAFT` models to perform lightweight solutions for problems where little or no data from the target domain is available.

On the other hand, there are certain limitations of our method that the users may face and should keep in mind. Firstly, the performance of these methods is highly dependent on the availability of DAFT models. `DAFT-E` or `DAFT-E`$^Z$ will not perform well when none of the available `DAFT` models are adjacent to the target domain. Moreover, while many `DAFT` models for sentiment analysis and NLI tasks are currently available, `DAFT`s for some other tasks might not be as widely available soon. Secondly, `DAFT-E`$^Z$ and `DAFT-E` require inference with multiple `DAFT` models. While the multi-model inference can be parallelized, thereby reducing inference time in scenarios where multiple smaller servers are available, inference time would increase if memory is limited or the per-model inference needs to be sequential. Further, with DAFT-E, we can filter out some of the candidate DAFT models, thereby reducing $N$, but that depends on having some data from the target domain. Thirdly, with the proposed methods, the users are supposed to use openly available fine-tuned models; however, the quality or correctness of the fine-tuning datasets and training methods may be questionable. There is also the question of the privacy of the data and the use of the models, which the user might not be aware of. Lastly, the risk of bias and misuse of using the DAFT models is possible. However, in this work, we are not proposing new models, just some methodologies to use already available models through public repositories. Since we are not changing the DAFT models, if the fine-tuning of the DAFT models were done according to safety and bias alignment, we expect to have minimal risk of bias and misuse from using ensembles of these models.

### Acknowledgments

The work was supported by RPI-IBM Future of Computing Research Collaboration (https://fcrc.rpi.edu).

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

# A  Appendix

## A.1  Datasets and Models

The attributes of the six datasets used for sentiment analysis and the three dataset used for text similarity analysis are given in Table 7 and 8. For 'SST2' we had to use validation dataset as test

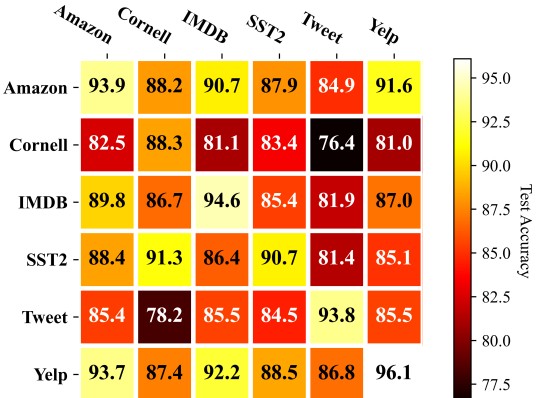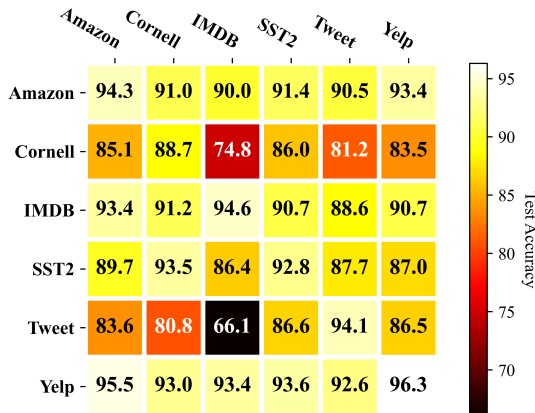

Figure 11: Heatmap showing the performance of DAFT models for sentiment analysis task when base model is BERT-based-uncased

Figure 12: Heatmap showing the performance of DAFT models for sentiment analysis task when base model is xlnet-base-cased

data, because the test data does not have any labels. The direct link to download these datasets are given as follows: https:// huggingface.co /datasets/amazon_polarity, 'https:// www.cs.cornell.edu /people/pabo/movie-review-data/', 'https://huggingface.co/datasets/ imdb', 'https:// huggingface.co/ datasets/sst2', 'https:// huggingface.co/ datasets/ mteb/tweet_sentiment_extraction', 'https:// huggingface.co/ datasets/ yelp_polarity', 'https://huggingface.co /datasets/nyu-mll/glue' (when copying the links please remove any spaces from the texts). Figs. 11 and 12 shows the performance heat-map for sentiment analysis task when the base models are BERT-based-uncased and xlnet-base-cased respectively. The heat maps in general follows a similar pattern as observed in Fig. 2, i.e., Tweet test data cannot be predicted well by any other DAFT models, and Cornell test data is also hard to predict. The observation of DAFT trained on tweet data performing poor is true for bert-based-uncased as well, but not always true for xlnet-base-cased. Another interesting observation is that for xlnet, the DAFT model trained on IMDB data seems to perform quite poorly in a few instances (i.e., on test data of Cornell and Tweet).

## A.2 Experimental Settings

The base models were used from huggingface using the 'AutoTokenizer.from_pretrained', and the webpage with descriptions and examples can be found in 'https://huggingface.co/ transformers/v3.0.2/model_doc/auto.html'. To fine-tune and train these models we used Google Colab platform with the T4 GPU equipped machine. For FT we chose each of the three base models (Roberta, BERT, xlnet) and fine-tuned the models with target data until the loss per epoch did not improve more than 1% or at least a fixed number of data-samples has not been used for training. All these fine-tuned models on the target dataset acted as a suitable candidate for DAFT models. For DA(FT)$^2$ we used any of these DAFT models to fine-tune on another target dataset for few shot training. In the performance comparison of DA(FT)$^2$ we only showed the performance of DA(FT)$^2$ that was fine-tuned on DAFT models having Roberta as its base. For both FT and DA(FT)$^2$ on few shot training, we performed the all the runs five times with five different seeds. For the case of DAFT-E, the weight calculations were done using five different random seeds as well. All the results shown here are the mean values of all those runs.

## A.3 Discussion on Performance Anomalies - SST2

The SST2 dataset obtained from Huggingface did not have any label on its test data, and we had to use the validation data of SST2 as the test data instead. Also, the validation data of SST2 only had 872 samples. Now, in the heat map shown in Fig. 2, the performance of DAFT-SST2 achieved 92% accuracy on the SST2 validation data, whereas, DAFT-Cornell achieved 93.1% accuracy on that same test dataset. This is quite counter intuitive, because we expect DAFT-SST2 to be the best performing one on its own domain (SST2

Table 7: Dataset sizes and classes (Sentiment Analysis)

| Dataset | Train data | Test data | Classes |
|---------|-----------|-----------|---------|
| Amazon | 3,600,000 | 400,000 | 2 |
| Cornell | 7,463 | 2134 | 2 |
| IMDB | 25,000 | 25,000 | 2 |
| SST2 | 67,349 | 1821 | 2 |
| Tweet | 12927 | 3696 | 3 |
| Yelp | 560,000 | 38,000 | 2 |

Table 8: Dataset sizes and classes (Text Similarity)

| Dataset | Train data | Test data | Classes |
|---------|-----------|-----------|---------|
| MRPC | 3,670 | 1,730 | 2 |
| QQP | 364,000 | 40,000 | 2 |
| STSB | 7,500 | 1,500 | 2 |

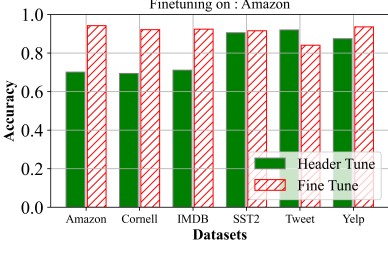

Figure 13: Performance comparison of FT Header layer to FT whole Model - Amazon dataset.

Figure 14: Performance comparison of FT Header layer to FT whole Model - Cornel dataset.

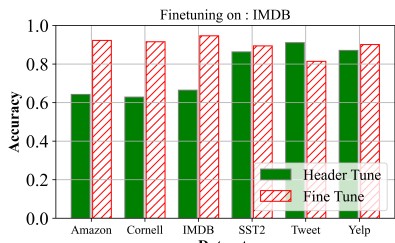

Figure 15: Performance comparison of FT Header layer to FT whole Model - IMDB dataset.

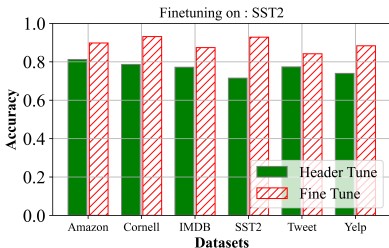

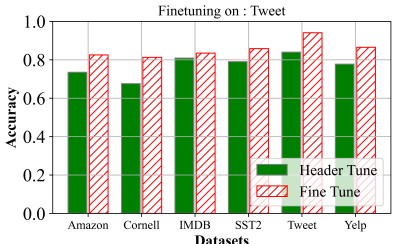

Figure 16: Performance comparison of FT Header layer to FT whole Model - SST2 dataset.

Figure 17: Performance comparison of FT Header layer to FT whole Model - Tweet dataset.

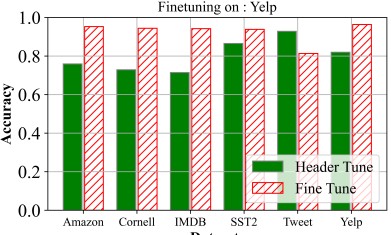

Figure 18: Performance comparison of FT Header layer to FT whole Model - Yelp dataset.

validation data). We did a thorough simulation analysis to check what might be the reason. Since the `DAFT`-SST2 was generated using randomly chosen $7,000$ samples from the training dataset of SST2, and we initially extracted the rest of the training data and divided it to different batches of $5,000$ samples. Now, with this batches we checked the performance of `DAFT`- SST2 and `DAFT`- Cornell. As expected, we got better performance with `DAFT`- SST2 compared to `DAFT`- Cornell with these new batches. With this experiment, we concluded that SST2 validation dataset is, for some unknown reason, more similar to the Cornell training dataset than the SST2 training data, and thus giving a counter intuitive result.

### A.4 Fine-tuning the header or whole model

Figs. 13 to 18 show the performance comparison of fine-tuning only the header layer of the base model and fine-tuning the whole model of the base models. It is straightforward to see the superior performance of fine-tuning the whole model even though the base models (i.e., BERT, Roberta) have already gone through extensive training with language data.

## A.5 Advantage of Ensemble

Our goal in this work is to find a suitable method that ensures better performance when data from the target domain is scarce, and therefore it is not possible to determine conclusively which of the available fine-tuned foundation models are adjacent to the target domain (if DAFT or not)[13]. The advantage of using Ensemble method are: 1) there is no constraint on the DAFT model size and architecture, 2) no need of knowing the weights of the DAFT models, just an API access to the DAFT models is adequate. In our study, we found that even if we do not know how domain-adjacent the available models are, their aggregate ensemble can still perform well (with zero or very limited training) as long as there are some models in the ensemble that are domain-adjacent (although we may not know which ones). Furthermore, as we have observed in the paper, the weighted ensemble of the DAFT models (DAFT-E) can perform similarly to or better than the best single (available) DAFT model. In summary, without adequate data it may be hard to determine which models are DAFT and which are not, and a very low-data low-resource ensembling method like DAFT-E$^Z$ or DAFT-E can give us good results.

## A.6 Ensemble Layer of DAFT-E

### A.6.1 Weight update of $LR$

Our ensemble technique uses the final layer output of the DAFT models as the input of the ensemble layer. When performing DAFT-E with $LR$, If $w_i$ is the weight of model $i$ in the Ensemble weight layer, then the $LR$ performs the following:

$$\min_{w_i \in \mathbf{w}} \sum_{(X,y) \in D_T} \ell\left(\left(\sum_i w_i M_i(X)\right), y\right) + R(\mathbf{w})$$

where, $\ell$ is the loss calculated using the output from the DAFT models $(M_i(X))$, to the ground truth $y$. Also, the term $R(\mathbf{w})$ can be used to regularize (or control) the number of models $(N)$ we want to use for our final ensemble. To keep the method simple, we did not use the regularization when implementing DAFT-E.

### A.6.2 Regression methods: $LR$ and $RF$

We observed the following in terms of performance of the two regression methods: (i) $LR$ has better performance and less variation in performance for smaller sample data, (ii) for higher sample data, i.e., $n>32$, $RF$ usually performs very similar to $LR$, (iii) for sentiment analysis tasks, $LR$ and $RF$ both perform well with similar performance at higher sample data, and (iv) for textual similarity tasks, $LR$ shows minimal performance improvement with more samples, whereas $RF$ shows promising results.

For $LR$ we have used SGDRegressor from sklearn.linear_model with maximum iteration of 3. The reason behind this small iteration number is because of the few shot regime. If we had used larger bound on iteration, overfitting might have casued more harm than good. We also used coefficient initialization $= 1/N$, where $N$ is the number of DAFT models used, so that at the start DAFT-E gives same weights to all the models. For $RF$ we imported the RandomForestClassifier from sklearn.ensemble, and

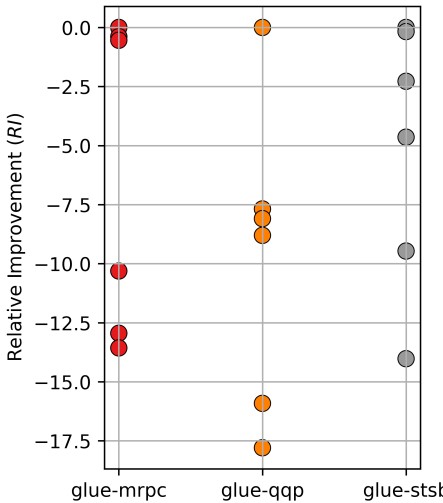

Figure 19: *Relative Improvement of DAFTs compared to the Single Best DAFT* for Text similarity task: For each test dataset, we consider 6 DAFT FMs (fine-tuned on data different from the test data), and consider the RI of DAFT$^Z$ over the single-best DAFT FM (out of the 6). Values less than 0 indicate performance degradation.

---

[13]There are a few established methods (e.g. LEEP score) to find the adjacency between two datasets, however these methods for determining domain-adjacency generally require a substantial amount of extra data.

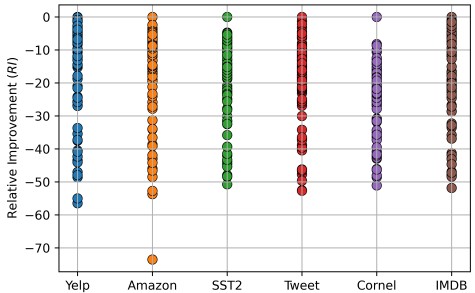 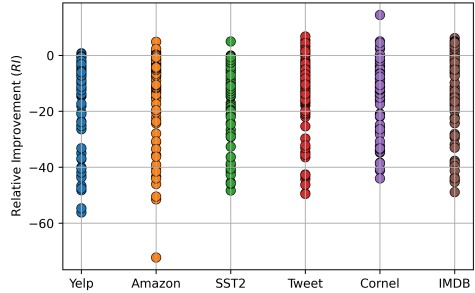

Figure 20: *Relative Improvement of DAFTs (from Huggingface) compared to the Single Best DAFT for sentiment analysis task. Values less than 0 indicate performance degradation.*

Figure 21: *Relative Improvement of DAFTs (from Huggingface) compared to DAFT-E $^Z$ for sentiment analysis task. Values less than 0 indicate performance degradation.*

set the max depth = 2. The smaller max depth was chosen to avoid overfitting.

### A.7    Performance of DAFT models for Textual similarity task

Fig. 19 shows the Relative Improvement of DAFTs compared to Single Best DAFT and DAFT-E$^Z$ respectively for the textual similarity task with three datasets. Similar to the results with sentiment analysis task, we observe the performance deviation among the DAFT models and usual better performance of DAFT-E$^Z$.

### A.8    Performance of DAFT models from Internet

To evaluate the performance of the readily available DAFT models from public repositories, we have downloaded 100 sentiment analysis models from Huggingface using 'popularity' as the sorting criterion. The performance of these 100 models compared to the single best DAFT and DAFT-E $^Z$ is shown in Figs. 20 and 21 respectively. Similar to what we observed previously, DAFT-E$^Z$ still outperforms most of the DAFT performances (551 among 600 cases). Moreover, it is notable that a lot of the DAFT models are very poor performing compared to our controlled DAFT models introduced in Section 3.

The performance of DAFT-E for the 100 models from Huggingface is depicted in Fig. 22. The results are consistent with the results that we observed in Fig. 8. In all cases, there is an upward trend in performance w.r.t. training data with the use of DAFT-E. Also, for 3 of the 6 datasets, DAFT-E catches up (or surpasses) the single best DAFT with 128 training pairs. Since we do not have the FFT for this scenario, the FFT comparison is not shown in Fig. 22.

### A.9    Proof of Propositions

**Proof of Proposition 1**    In the following, we let $\mathcal{N} = \{1, 2, \cdots, N\}$ denote the set of indices of the DAFT models in the ensemble.

$$
\begin{aligned}
\mathbb{E}(\ell(\mathsf{DAFT}^{\mathsf{Z}})) &= \mathbb{E}(\ell(M_i(X), y)) \quad \forall i \in \mathcal{N} \\
&= \frac{1}{N} \sum_{i \in \mathcal{N}} \ell(M_i(X), y) \\
&= \frac{1}{N} \sum_{i \in \mathcal{N}} \ell(\mu_i, y) \quad [\text{denoting } M_i(X) \text{ as } \mu_i] \\
&\geq \ell\left(\sum_{i \in \mathcal{N}} \frac{1}{N}\mu_i, y\right) \quad [\text{assuming } \ell \text{ is convex in } \mu_i] \\
&= \text{loss of average ensemble} = \ell(\mathsf{DAFT\text{-}E}^{\mathsf{Z}}).
\end{aligned} \tag{2}
$$

Note that the derivation assumes $\ell$ to be convex in $\mu_i$, the outputs from the DAFT models. We assume a convex loss function since the loss is calculated using the output probabilities of each DAFT model, which

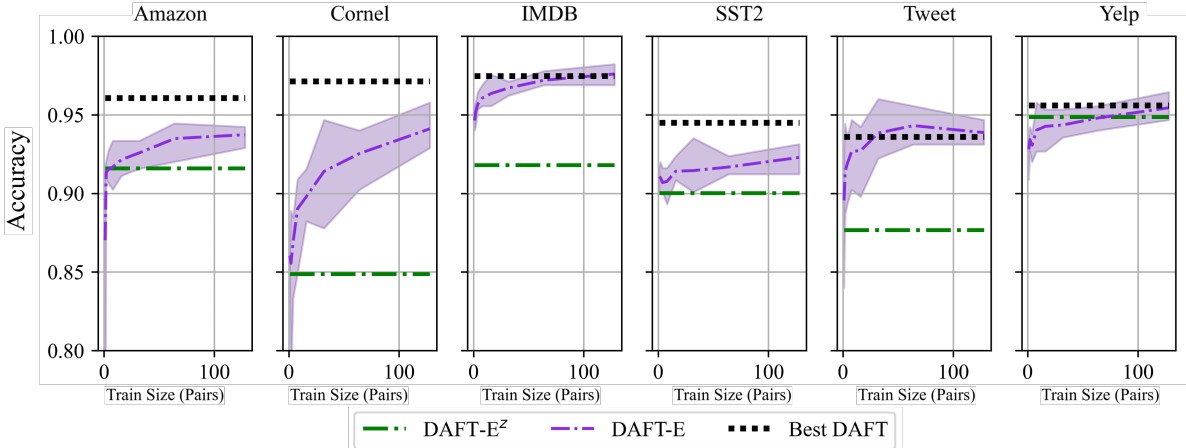

Figure 22: *Performance comparison of DAFT-E$^Z$, DAFT-E, and single-best.* Error interval for DAFT-E is based on the random choice of the few-shot samples used to learn the weights of the ensemble aggregated over 10 trials.

is a continuous value. Also, the most popular loss functions like MAE, or MSE are directly proportional to the distance between the predicted output $\hat{y}_{\text{pred}}$ and the true output $y$, and are convex functions (Terven et al., 2023). □

**Proof of Proposition 2** If the ensemble weights for the $i^{th}$ model is $w_i$, then for the weighted ensemble method we have;

$$
\begin{aligned}
&\ell(\text{DAFT-E}) \\
&= \text{loss of weighted ensemble} \\
&= \min_{w_i, i \in \mathcal{N}} \left[ \ell \left( \sum_i w_i M_i \right) \right] \\
&\leq \ell(M_i), \;\; \forall i \in \mathcal{N}.
\end{aligned}
\tag{3}
$$

Note that the above derivation assumes that the set weights $w_i, i \in \mathcal{N}$, have been optimized (trained using the fine-tuning dataset from the target domain) to minimize the loss function $\ell \left( \sum_i w_i M_i \right)$.

Note that the base model that $M_i$ is built from is $B_{\kappa(i)}$. Further, $\tilde{M}_i$ denotes the corresponding FFT, i.e., the model built from $B_{\kappa(i)}$ by fine-tuning on the target dataset $D_T$. Now, let us assume that for two DAFT models $M_1$ and $M_2$ developed from the same base model and fine-tuned on different datasets, i.e., $D_{\nu(1)}$ and $D_{\nu(2)}$, to have the following bound on their losses;

$$
|\ell(M_1) - \ell(M_2)| \leq \rho(D_{\nu(1)}, D_{\nu(2)}),
\tag{4}
$$

where $\rho$ is an appropriately defined distance measure between the datasets $D_{\nu(1)}$ and $D_{\nu(2)}$. Then from Equation 3 we have;

$$
\ell(\text{DAFT-E}) \leq \ell(M_i) \leq \ell(\tilde{M}_i) + \rho(D_{\nu(i)}, D_T), \forall i.
\tag{5}
$$

Now, let us denote $\tilde{M}_*$ be the model fine-tuned on $D_T$ and performs the best among all the models when fine-tuned on $D_T$ (all possible FFTs). Then from 5 and by defining $\mu(M_1, M_2) \overset{\Delta}{=} \ell(M_1) - \ell(M_2)$, we have

$$
\begin{aligned}
\ell(\text{DAFT-E}) \leq \ell(\tilde{M}_*) &+ \rho(D_{\nu(i)}, D_T) \\
&+ \mu(\tilde{M}_i, \tilde{M}_*); \;\; \forall i.
\end{aligned}
\tag{6}
$$

Since Equation 6 holds for all $i \in \mathcal{N}$, Equation 1 follows by taking a minimum over all $i$, completing the proof. □

Note that Equation 1 bounds the gap between the mimimum loss possible under any model and the loss of DAFT-E. The result is intuitive. The term $\mu()$ tells us that the performance gap depends on how good the base models (corresponding to the DAFT models in the ensemble) are, when compared to the best possible model, when they are all trained on the full target dataset. The term $\rho()$ implies that the performance gap also depends on how closely the datasets used to generate the DAFT models represent the target dataset. Note that the minimum over $i$ is taken on the sum of $\mu()$ and $\rho()$, instead of of each of the two terms individually. However, for an ensemble of DAFT models obtained by fine-tuning a "good" set of FMs with a large number of datasets, the approximation term $\mu() + \rho()$ is expected to be small, as argued in Section 5.2.

Table 9: Relative Improvement of $\mathtt{DA(FT)}^2$ compared to $\mathtt{FT}$ (Sentiment Analysis). The base model for all the models ($\mathtt{DAFT}$ and $\mathtt{FT}$) here is Roberta-base.

| Dataset | DAFT Name | $n=2$ | $n=4$ | $n=8$ | $n=16$ | $n=32$ | $n=64$ | $n=128$ | $n=256$ |
|---|---|---|---|---|---|---|---|---|---|
| Amazon | DAFT- Cornell | 80.42 | 76.28 | 81.10 | 67.16 | 55.45 | 32.55 | 0.50 | 0.48 |
| | DAFT- IMDB | 78.33 | 77.79 | 83.46 | 68.53 | 57.12 | 37.85 | 2.66 | 1.92 |
| | DAFT- SST2 | 78.11 | 75.33 | 78.67 | 65.6 | 54.45 | 34.45 | 0.57 | 0.77 |
| | DAFT- Tweet | 61.33 | 67.70 | 71.34 | 56.76 | 47.07 | 31.19 | -1.11 | -0.85 |
| | DAFT- Yelp | 81.59 | 79.18 | 83.50 | 68.95 | 53.48 | 35.24 | 1.42 | 1.54 |
| | Average | 75.95 | 75.25 | 79.62 | 65.40 | 53.51 | 34.26 | 0.81 | 0.77 |
| Cornell | DAFT- Amazon | 73.60 | 71.74 | 68.88 | 63.6 | 61.01 | 38.06 | 10.25 | 1.66 |
| | DAFT- IMDB | 68.43 | 64.88 | 64.64 | 62.43 | 61.23 | 38.52 | 9.58 | 3.60 |
| | DAFT- SST2 | 74.8 | 72.11 | 68.89 | 66.71 | 63.54 | 38.97 | 9.64 | 2.67 |
| | DAFT- Tweet | 53.72 | 52.43 | 49.77 | 48.37 | 51.15 | 31.50 | 6.11 | 2.23 |
| | DAFT- Yelp | 69.22 | 66.87 | 64.13 | 62.47 | 51.87 | 32.19 | 8.65 | 0.3 |
| | Average | 67.95 | 65.61 | 63.26 | 60.72 | 57.77 | 35.85 | 8.85 | 2.09 |
| IMDB | DAFT- Amazon | 81.35 | 81.84 | 81.66 | 70.94 | 61.86 | 50.11 | 2.53 | 0.65 |
| | DAFT- Cornell | 77.79 | 77.09 | 80.67 | 66.68 | 57.4 | 49.72 | 2.46 | -0.03 |
| | DAFT- SST2 | 76.10 | 75.28 | 77.99 | 64.86 | 57.45 | 47.36 | -1.33 | -0.977 |
| | DAFT- Tweet | 59.12 | 60.27 | 60.49 | 58.10 | 53.54 | 45.20 | -0.62 | -0.70 |
| | DAFT- Yelp | 76.93 | 78.72 | 80.11 | 69.47 | 60.73 | 45.82 | -2.52 | 0.247 |
| | Average | 74.26 | 74.64 | 76.19 | 66.01 | 58.20 | 47.64 | 0.10 | -0.16 |
| SST2 | DAFT- Amazon | 76.35 | 73.84 | 73.54 | 70.05 | 72.62 | 58.52 | 11.47 | 2.03 |
| | DAFT- Cornell | 82.05 | 78.97 | 78.81 | 80.32 | 77.56 | 64.29 | 16.84 | 5.41 |
| | DAFT- IMDB | 71.55 | 75.20 | 72.81 | 74.51 | 71.62 | 62.14 | 14.46 | 3.85 |
| | DAFT- Tweet | 62.25 | 60.66 | 60.68 | 62.66 | 63.46 | 53.84 | 10.30 | 0.92 |
| | DAFT- Yelp | 73.42 | 74.51 | 73.2 | 72.18 | 74.17 | 54.1 | 10.89 | 2.75 |
| | Average | 73.13 | 72.64 | 71.81 | 71.95 | 71.88 | 58.58 | 12.79 | 2.99 |
| Tweet | DAFT- Amazon | 60.22 | 53.93 | 55.79 | 41.88 | 42.67 | 26.16 | 1.2 | -2.2 |
| | DAFT- Cornell | 56.37 | 49.64 | 50.94 | 52.62 | 39.55 | 23.64 | 1.03 | -1.58 |
| | DAFT- IMDB | 59.72 | 53.76 | 52.58 | 55.26 | 45.22 | 30.19 | 4.69 | 1.59 |
| | DAFT- SST2 | 62.23 | 55.20 | 57.46 | 54.1 | 41.79 | 26.48 | 2.29 | -1.06 |
| | DAFT- Yelp | 64.83 | 58.69 | 60.85 | 57.37 | 39.85 | 23.15 | 2.5 | -1.6 |
| | Average | 60.67 | 54.25 | 55.53 | 52.25 | 41.82 | 25.92 | 2.34 | -0.97 |
| Yelp | DAFT- Amazon | 79.81 | 84.56 | 71.46 | 85.73 | 49.85 | 26.89 | -2.32 | -0.41 |
| | DAFT- Cornell | 74.39 | 83.52 | 67.09 | 81.15 | 45.62 | 29.28 | -0.65 | -0.81 |
| | DAFT- IMDB | 77.45 | 83.51 | 67.84 | 83.75 | 47.97 | 29.62 | 1.02 | 0.29 |
| | DAFT- SST2 | 75.4 | 80.74 | 66.07 | 80.69 | 45.82 | 26.711 | -0.86 | -1.48 |
| | DAFT- Tweet | 67.36 | 72.68 | 61.60 | 77.06 | 43.10 | 26.81 | -0.79 | -1.22 |
| | Average | 74.88 | 81.00 | 66.81 | 81.68 | 46.47 | 27.86 | -0.72 | -0.73 |

Table 10: Relative Improvement of $\text{DA}(\text{FT})^2$ compared to FT (Text Similarity). The R, B, X at the end of the DAFT names denote the base model of the DAFT models, and they denote: Roberta, Bert, Xlnet base models respectively.

| Dataset | DAFT Name | $n=2$ | $n=4$ | $n=8$ | $n=16$ | $n=32$ | $n=64$ | $n=128$ | $n=256$ |
|---|---|---|---|---|---|---|---|---|---|
| MRPC | DAFT- QQP-R | 34.78 | 32.60 | 32.08 | -6.49 | 24.68 | 18.15 | 20.21 | 10.83 |
| | DAFT- STSB-R | 55.83 | 54.79 | 61.86 | 13.01 | 41.26 | 32.65 | 28.13 | 18.20 |
| | DAFT- QQP-B | 36.45 | 34.58 | 39.62 | -1.60 | 24.19 | 16.03 | 12.41 | 8.69 |
| | DAFT- STSB-B | 54.27 | 52.31 | 59.27 | 12.49 | 37.97 | 27.81 | 23.47 | 12.33 |
| | DAFT- QQP-X | 33.75 | 29.95 | 35.87 | -2.57 | 24.02 | 16.12 | 15.38 | 6.05 |
| | DAFT- STSB-X | 54.22 | 53.29 | 59.77 | 13.61 | 40.77 | 30.08 | 25.80 | 16.34 |
| | Average | 44.88 | 42.92 | 48.08 | 4.74 | 32.15 | 23.47 | 20.90 | 12.07 |
| QQP | DAFT- MRPC-R | 75.56 | 78.10 | 88.81 | 86.98 | 66.05 | 47.51 | 16.33 | 4.71 |
| | DAFT- STSB-R | 81.24 | 80.61 | 99.21 | 97.98 | 77.47 | 57.41 | 19.36 | 7.38 |
| | DAFT- MRPC-B | 63.40 | 65.81 | 72.71 | 78.10 | 59.59 | 42.61 | 9.76 | -0.62 |
| | DAFT- STSB-B | 79.18 | 82.73 | 89.69 | 94.59 | 69.40 | 51.00 | 14.10 | 2.03 |
| | DAFT- MRPC-X | 61.59 | 65.76 | 72.62 | 75.59 | 56.21 | 42.88 | 8.57 | -2.41 |
| | DAFT- STSB-X | 76.61 | 78.03 | 83.61 | 88.56 | 63.45 | 49.58 | 15.04 | 1.97 |
| | Average | 72.93 | 75.17 | 84.44 | 86.97 | 65.36 | 48.50 | 13.86 | 2.18 |
| STSB | DAFT- MRPC-R | 46.89 | 53.02 | 50.07 | 51.96 | 45.97 | 28.01 | 1.26 | 0.64 |
| | DAFT- QQP-R | 51.78 | 50.82 | 57.42 | 63.87 | 56.15 | 36.55 | 4.85 | 1.95 |
| | DAFT- MRPC-B | 42.88 | 40.53 | 44.46 | 47.67 | 44.77 | 24.74 | -3.51 | -3.24 |
| | DAFT- QQP-B | 56.55 | 58.26 | 56.90 | 58.49 | 57.51 | 33.36 | 1.83 | 0.89 |
| | DAFT- MRPC-X | 40.73 | 39.28 | 35.59 | 41.34 | 41.23 | 22.05 | -5.52 | -3.76 |
| | DAFT- QQP-X | 51.34 | 53.98 | 52.69 | 55.15 | 53.66 | 30.73 | 0.37 | -0.74 |
| | Average | 48.36 | 49.31 | 49.52 | 53.08 | 49.88 | 29.24 | -0.12 | -0.71 |

Table 11: Relative Improvement of $\mathsf{DA(FT)}^2$ compared to $\mathsf{DAFT\text{-}E}$ (Sentiment Analysis). The base model for all the $\mathsf{DAFT}$ models here is Roberta-base.

| Dataset | DAFT Name | $n=2$ | $n=4$ | $n=8$ | $n=16$ | $n=32$ | $n=64$ | $n=128$ | $n=256$ |
|---|---|---|---|---|---|---|---|---|---|
| Amazon | DAFT- Cornell | -1.60 | -1.91 | -1.57 | -1.38 | -1.43 | -4.177 | -2.35 | -1.78 |
| | DAFT- IMDB | -2.74 | -1.07 | -0.29 | -0.57 | -0.38 | -0.35 | -0.26 | -0.37 |
| | DAFT- SST2 | -2.85 | -2.43 | -2.89 | -2.3 | -2.06 | -2.80 | -2.30 | -1.50 |
| | DAFT- Tweet | -12.01 | -6.68 | -6.87 | -7.51 | -6.75 | -5.17 | -3.92 | -3.09 |
| | DAFT- Yelp | -0.96 | -0.30 | -0.27 | -0.32 | -2.683 | -2.238 | -1.46 | -0.753 |
| | Average | -4.03 | -2.48 | -2.38 | -2.42 | -2.66 | -2.95 | -2.06 | -1.50 |
| Cornell | DAFT- Amazon | -0.71 | -0.03 | -0.04 | -1.81 | -2.14 | -2.34 | -1.27 | -3.06 |
| | DAFT- IMDB | -3.66 | -4.02 | -2.55 | -2.51 | -2.03 | -2.02 | -1.87 | -1.21 |
| | DAFT- SST2 | -0.02 | 0.187 | -0.04 | 0.05 | -0.62 | -1.70 | -1.82 | -2.10 |
| | DAFT- Tweet | -12.08 | -11.27 | -11.35 | -10.95 | -8.15 | -6.99 | -4.98 | -2.52 |
| | DAFT- Yelp | -3.21 | -2.87 | -2.85 | -2.49 | -7.72 | -6.50 | -2.70 | -4.38 |
| | Average | -3.94 | -3.60 | -3.37 | -3.54 | -4.13 | -3.91 | -2.53 | -2.65 |
| IMDB | DAFT- Amazon | 1.25 | 0.57 | -0.35 | -0.43 | -0.79 | -1.25 | -0.48 | -0.65 |
| | DAFT- Cornell | -0.73 | -2.05 | -0.89 | -2.92 | -3.53 | -1.51 | -0.54 | -1.32 |
| | DAFT- SST2 | -1.675 | -3.05 | -2.36 | -3.98 | -3.50 | -3.06 | -4.22 | -2.25 |
| | DAFT- Tweet | -11.16 | -11.35 | -11.96 | -7.92 | -5.89 | -4.48 | -3.53 | -1.98 |
| | DAFT- Yelp | -1.21 | -1.25 | -1.20 | -1.29 | -1.49 | -4.07 | -5.39 | -1.05 |
| | Average | -2.70 | -3.41 | -3.35 | -3.31 | -3.04 | -2.87 | -2.83 | -1.45 |
| SST2 | DAFT- Amazon | -2.89 | -2.64 | -2.94 | -5.91 | -3.70 | -4.71 | -5.33 | -4.65 |
| | DAFT- Cornell | 0.25 | 0.23 | -0.02 | -0.24 | -0.95 | -1.25 | -0.77 | -1.49 |
| | DAFT- IMDB | -5.53 | -1.88 | -3.38 | -3.45 | -4.26 | -2.54 | -2.79 | -2.95 |
| | DAFT- Tweet | -10.65 | -10.03 | -10.16 | -10.01 | -8.81 | -7.53 | -6.33 | -5.69 |
| | DAFT- Yelp | -4.50 | -2.27 | -3.16 | -4.74 | -2.84 | -7.37 | -5.82 | -3.97 |
| | Average | -4.67 | -3.32 | -3.94 | -4.87 | -4.11 | -4.68 | -4.21 | -3.75 |
| Tweet | DAFT- Amazon | -4.89 | -5.09 | -4.98 | -11.98 | -3.42 | -3.64 | -2.69 | -1.63 |
| | DAFT- Cornell | -7.17 | -7.74 | -7.94 | -5.31 | -5.53 | -5.56 | -2.86 | -1.00 |
| | DAFT- IMDB | -5.19 | -5.21 | -6.93 | -3.68 | -1.69 | -0.57 | 0.67 | 2.18 |
| | DAFT- SST2 | -3.70 | -4.31 | -3.96 | - 4.40 | -4.02 | -3.40 | -1.64 | -0.48 |
| | DAFT- Yelp | -2.15 | -2.17 | -1.89 | -2.36 | -5.33 | -5.94 | -1.44 | -1.02 |
| | Average | -4.62 | -4.91 | -5.14 | -5.54 | -4.00 | -3.82 | -1.59 | -0.39 |
| Yelp | DAFT- Amazon | 0.18 | -0.84 | 0.62 | 0.10 | 0.60 | -3.20 | -3.85 | -0.98 |
| | DAFT- Cornell | -2.83 | -1.40 | -1.93 | -2.37 | -2.25 | -1.37 | -2.21 | -1.37 |
| | DAFT- IMDB | -1.13 | -1.41 | -1.50 | -0.97 | -0.67 | -1.11 | -0.57 | -0.29 |
| | DAFT- SST2 | -2.27 | -2.90 | -2.53 | -2.62 | -2.11 | -3.33 | -2.42 | -2.04 |
| | DAFT- Tweet | -6.75 | -7.22 | -5.16 | -4.58 | -3.94 | -3.26 | -2.34 | -1.79 |
| | Average | -2.56 | -2.76 | -2.10 | -2.09 | -1.68 | -2.46 | -2.28 | -1.29 |

Table 12: Relative Improvement of $\mathsf{DA(FT)}^2$ compared to $\mathsf{DAFT\text{-}E\text{-}}$ LR (Text Similarity). The R, B, X at the end of the $\mathsf{DAFT}$ names denote the base model of the $\mathsf{DAFT}$ models, and they denote: Roberta, Bert, Xlnet base models respectively.

| Dataset | DAFT Name | $n=2$ | $n=4$ | $n=8$ | $n=16$ | $n=32$ | $n=64$ | $n=128$ | $n=256$ |
|---|---|---|---|---|---|---|---|---|---|
| MRPC | DAFT- QQP-R | -11.80 | -11.25 | -14.47 | -13.93 | -8.60 | -2.85 | 4.15 | 7.20 |
| | DAFT- STSB-R | 1.98 | 3.60 | 4.81 | 4.01 | 3.56 | 9.08 | 11.00 | 14.33 |
| | DAFT- QQP-B | -10.71 | -9.92 | -9.59 | -9.44 | -8.95 | -4.59 | -2.61 | 5.13 |
| | DAFT- STSB | 0.95 | 1.94 | 3.14 | 3.53 | 1.15 | 5.10 | 6.97 | 8.66 |
| | DAFT- QQP-X | -12.47 | -13.02 | -12.02 | -10.33 | -9.08 | -4.52 | -0.04 | 2.58 |
| | DAFT- STSB-X | 0.92 | 2.60 | 3.46 | 4.56 | 3.20 | 6.96 | 8.99 | 12.54 |
| | Average | -5.19 | -4.34 | -4.11 | -3.60 | -3.12 | 1.53 | 4.74 | 8.41 |
| QQP | DAFT- MRPC-R | -1.37 | -1.01 | 1.90 | -0.05 | 2.44 | 1.73 | 6.68 | 7.89 |
| | DAFT- STSB-R | 1.82 | 0.39 | 7.51 | 5.83 | 9.49 | 8.55 | 9.46 | 10.65 |
| | DAFT- MRPC-B | -8.20 | -7.84 | -6.80 | -4.80 | -1.55 | -1.65 | 0.65 | 2.40 |
| | DAFT- STSB-B | 0.66 | 1.57 | 2.37 | 4.01 | 4.51 | 4.13 | 4.64 | 5.13 |
| | DAFT- MRPC-X | -9.22 | -7.87 | -6.84 | -6.14 | -3.63 | -1.46 | -0.43 | 0.56 |
| | DAFT- STSB-X | -0.78 | -1.05 | -0.91 | 0.79 | 0.83 | 3.16 | 5.50 | 5.07 |
| | Average | -2.85 | -2.64 | -0.46 | -0.06 | 2.01 | 2.41 | 4.42 | 5.28 |
| STSB | DAFT- MRPC-R | -6.23 | -3.43 | -3.72 | -2.61 | -4.53 | -0.39 | 2.68 | 4.96 |
| | DAFT- QQP-R | -3.11 | -4.82 | 0.99 | 5.03 | 2.13 | 6.25 | 6.32 | 6.32 |
| | DAFT- MRPC-B | -8.79 | -11.31 | -7.32 | -5.35 | -5.32 | -2.93 | -2.15 | 0.91 |
| | DAFT- QQP-B | -0.07 | -0.12 | 0.66 | 1.58 | 3.02 | 3.77 | 3.26 | 5.21 |
| | DAFT- MRPC-X | -10.16 | -12.10 | -13.01 | -9.41 | -7.63 | -5.03 | -4.20 | 0.36 |
| | DAFT- QQP-X | -3.39 | -2.82 | -2.04 | -0.56 | 0.50 | 1.73 | 1.78 | 3.51 |
| | Average | -5.29 | -5.77 | -4.07 | -1.89 | -1.97 | 0.57 | 1.28 | 3.55 |

Table 13: Relative Improvement of $\mathsf{DA(FT)}^2$ compared to $\mathsf{DAFT\text{-}E\text{-}}$ RF (Text Similarity). The R, B, X at the end of the $\mathsf{DAFT}$ names denote the base model of the $\mathsf{DAFT}$ models, and they denote: Roberta, Bert, Xlnet base models respectively.

| Dataset | DAFT Name | $n=2$ | $n=4$ | $n=8$ | $n=16$ | $n=32$ | $n=64$ | $n=128$ | $n=256$ |
|---|---|---|---|---|---|---|---|---|---|
| MRPC | DAFT- QQP-R | 12.91 | -2.02 | -10.60 | -14.67 | -11.38 | -7.66 | -3.32 | -0.07 |
| | DAFT- STSB-R | 30.55 | 14.37 | 9.56 | 3.12 | 0.40 | 3.67 | 3.04 | 6.58 |
| | DAFT- QQP-B | 14.31 | -0.56 | -5.49 | -10.21 | -11.73 | -9.32 | -9.59 | -1.99 |
| | DAFT- STSB | 29.24 | 12.54 | 7.81 | 2.65 | -1.94 | -0.11 | -0.70 | 1.29 |
| | DAFT- QQP-X | 12.05 | -3.98 | -8.03 | -11.09 | -11.86 | -9.25 | -7.21 | -4.38 |
| | DAFT- STSB-X | 29.20 | 13.26 | 8.15 | 3.67 | 0.05 | 1.66 | 1.17 | 4.91 |
| | Average | 28.02 | 10.39 | 4.53 | -0.85 | -3.05 | -4.96 | -7.17 | -6.78 |
| QQP | DAFT- MRPC-R | 5.92 | 2.20 | 2.44 | -0.88 | -1.49 | -3.20 | 1.52 | 3.59 |
| | DAFT- STSB-R | 9.35 | 3.64 | 8.08 | 4.95 | 5.29 | 3.29 | 4.17 | 6.23 |
| | DAFT- MRPC-B | -1.42 | -4.85 | -6.29 | -5.59 | -5.32 | -6.42 | -4.21 | -1.69 |
| | DAFT- STSB-B | 8.10 | 4.86 | 2.92 | 3.15 | 0.50 | -0.91 | -0.42 | 0.93 |
| | DAFT- MRPC-X | -2.51 | -4.88 | -6.34 | -6.92 | -7.32 | -6.24 | -5.25 | -3.46 |
| | DAFT- STSB-X | 6.56 | 2.16 | -0.38 | -0.04 | -3.03 | -1.84 | 0.40 | 0.88 |
| | Average | 7.39 | 3.24 | 0.54 | -0.83 | -3.83 | -4.85 | -4.83 | -3.99 |
| STSB | DAFT- MRPC-R | 11.08 | 5.44 | -4.39 | -5.82 | -8.44 | -6.09 | -2.41 | -0.63 |
| | DAFT- QQP-R | 14.77 | 3.92 | 0.29 | 1.56 | -2.06 | 0.18 | 1.05 | 0.66 |
| | DAFT- MRPC-B | 8.05 | -3.17 | -7.96 | -8.48 | -9.20 | -8.49 | -7.00 | -4.47 |
| | DAFT- QQP-B | 18.38 | 9.05 | -0.04 | -1.78 | -1.21 | -2.17 | -1.86 | -0.39 |
| | DAFT- MRPC-X | 6.42 | -4.03 | -13.61 | -12.40 | -11.42 | -10.46 | -8.94 | -4.99 |
| | DAFT- QQP-X | 14.44 | 6.10 | -2.72 | -3.85 | -3.62 | -4.09 | -3.26 | -2.00 |
| | Average | 18.46 | 9.18 | -0.69 | -3.30 | -4.10 | -5.72 | -4.95 | -5.33 |

