# OpenReview forum: "On the Utility of Existing Fine-Tuned Models on Data-Scarce Domains"
_TMLR — Accepted by TMLR_

### Review · Reviewer_NUGp · 2025-03-06

**Summary Of Contributions:**

The paper focuses on leveraging existing fine-tuned LLMs for the target tasks/domains where data are limited or not available (Note that the target tasks are similar to the source tasks where the LLMs are fine-tuned on). The authors studied several ways of utilizing a set of DAFTs for the target task under zero-shot or few-shot, including FT, DAFT^Z, DA(FT)^2, DAFT-E^Z and DAFT-E. And the authors found that ensembling of DAFTs achieves similar results to single best model in zero-shot settings. In few-shot problems, picking or putting more weights to the DAFT models that are expected to perform better on the target task can improve the performance.

**Audience:**

Yes

**Broader Impact Concerns:**

The paper included a statement of Broader Impact. Additional to that, I would recommend the authors to also consider the risk of bias when using the DAFT models and mitigation strategies as these model can bias to the data and lead to bias in real-world application. The authors are also recommended to include a discussion of potential misuse risks of DAFT models and safeguards against them.

**Claims And Evidence:**

Yes

**Requested Changes:**

* I would recommend the authors to **consider more general settings** where those DAFTs are from internet and there can be some overlap between the training and testing datasets. (See 1 in weaknesses)

* More **clearly explain the learning strategies** such as those mentioned in 2 in weaknesses.

* Considering **more transferability metrics [A]**, more recent model merging methods, e.g., [B] and discussion with some related work [C][D][E] that is missing in this paper

* **More discussion and definition of the DAFT**. And how to measure the similarity between tasks and datasets.

* Writings: I would recommend the authors to better proofread the paper. For example, in the first paragraph in the introduction, the authors claim that 'Usually, base models do not need much training to perform well, because the pre-trained LLMs are already trained on a large corpus of data.' And then they mentioned that 'To perform a task with base models, we can fine- tune a base model with task-specific training data, and then use the fine-tuned model.', which are not very consistent.

[A] Agostinelli et al., Transferability Metrics for Selecting Source Model Ensembles, CVPR, 2022

[B] Yang et al., AdaMerging: Adaptive Model Merging for Multi-Task Learning, ICLR, 2024

[C] Mensink et al., Factors of Influence for Transfer Learning across Diverse Appearance Domains and Task Types, PAMI, 2022

[D] Zamir et al., Taskonomy: Disentangling Task Transfer Learning, CVPR, 2018

[E] Dvornik et al., Selecting Relevant Features from a Multi-domain Representation for Few-shot Classification, ECCV, 2020

**Strengths And Weaknesses:**

+ ***Strengths***

1. In general, this is a very **practical problem** as it can **significantly save computational and memory cost** during the deployment of models for real-world applications.

2. The paper can be very helpful for studying the transferability between tasks and datasets.

- ***Weaknesses***

1 The motivation of the paper is that there can be **a large number of fine-tuned models** and can be used for the target task instead of considering the computationally intensive fully fine-tuning method. However, the experiments were conducted on **few fine-tune models** and would **limit their transferability to the general settings where there are thousands of DAFT models and overlaps between data of source and target tasks/datasets**.

2. It is not very clear how the base models are achieved. Are they obtained by fine-tuning the whole model or only learn the header layer?

3. While it is clear that one or some DAFT models can perform better than the best single model etc. However, as shown in Figure 2 (a) that the transferability between any pairs of tasks are **not consistent** and **how to pick the best model(s)**. Also **how to learn the weight to ensemble DAFT models are not clear as well**.

4. The work only leverage LEEP scores and I would recommend also consider analyze **other transferability metrics [A]** and also more general settings: getting (large number of) fine-tuned models from the internet.

5. Given that if we use the ensembles of thousands of fine-tuned DAFT models which requires forwarding a sample through thousands of models. **Would this cost more than fine-tune one model for the task and we only need forward once for the task?**

6. The authors defined DAFT as 'We call a model a domain adjacent fine-tuned model (DAFT) if it can be used for the same (general) task as the target dataset, i.e., the model was fine-tuned with a dataset that have the same task criterion as the target data.' However, in practice, **how to measure the similarity between tasks and datasets**?

[A] Agostinelli et al., Transferability Metrics for Selecting Source Model Ensembles, CVPR, 2022

---

> ### Author Response · Authors · 2025-03-18
> **Response to Requested Changes and Weaknesses**
>
> ### Dear Reviewer, thank you for your insightful comments and suggestions. We have made every effort to address them thoroughly as follows. We kindly invite your feedback on our responses.
> ### **Requested Changes**
> **RC1:** "I would recommend .. "
>
> *Response:* Thank you for your suggestion. We mainly used our own DAFT models to remove any data leakage issue; otherwise, our results might have been more favorable to DAFT-E. However, we agree with your recommendation to consider more general settings. We have searched for ’sentiment analysis’ and ’text similarity / NLI’ models in Huggingface and have made a list of hundreds of models. We plan to pick some of those models randomly and check the performance of the ensemble methods over a few iterations. We should be able to report/include some of these results within the revision timeframe.
>
> **RC2:** "More clearly explain..."
>
> *Response:* The learning strategies are discussed in footnote 4 on page 6. Alongside the footnote, we can add a line explicitly (to the main body) saying how the DAFT models were fine-tuned.
>
> **RC3:** "Considering more transferability ..."
>
> *Response:* Thank you very much for suggesting these related works. We have read the papers and agree that adding these will improve the quality of the manuscript. More explicitly, we will add works of [C, D, E] in the Related Work section. For transferability metrics [A], the metric is developed on LEEP and the best performing one is Sof-IoU-EEP. We are implementing that method now and hope to report some results within this discussion/revision timeframe. Lastly, we will add the AdaMerging method to Table 1 to compare with other ensemble methods.
>
> **RC4:** "More discussion and....
>
> *Response:* We will formally define DAFT as a definition in the paper along with the conditions that it needs to meet. The task similarity can be calculated in two ways: (i) by the source and target dataset similarity and (ii) by task description similarity. Since in most cases the source dataset, with which the DAFT model was fine-tuned, is not available, method (ii) is more general. For method (ii), the task similarity can be identified manually or measured using an LLM fine-tuned for ‘textual similarity’. In our definition of DAFT, we are proposing to follow method (ii) to find DAFT models, i.e., when a model is fine-tuned with a dataset that has the same task criterion as the target data.
>
> For similarity measures, we appreciate the suggestion of using the methods from [B]. We have read the paper, and are planning to evaluate those metrics in our setup. However, we would like to emphasize that our focus is not on finding the optimum way to select source models for the ensemble, rather we are showing that even if limited data from the target domain is available, we can enhance the performance of the DAFT ensemble. In this context, it is also noteworthy that most transferability measures work subpar (from our observation with LEEP scores and other transferability measures) when the data availability from the target domain is minimal (data-scarce domain).
>
> **RC5:** "Writings: I would..."
>
> *Response:* Thank you. We will proofread the paper to check for consistency issues and take care to solve them.
>
> **RC-BIC:** "The paper included a statement of Broader Impact..."
>
> *Response:* Thank you, we will add the following in the Broader Impact Statement:
> The risk of bias and misuse of using the DAFT models is possible. However, in this work, we are not proposing new models, just some methodologies to use already available models through public repositories. Since we are not changing the DAFT models, if the fine-tuning of the DAFT models were done according to safety and bias alignment, we expect to have minimal risk of bias and misuse from using ensembles of these models.
>
> ### **Weakness**
> **Response to weakness 1, 2, 4 and 6 are made in requested changes.**
>
> **W3:** "While it is clear..."
>
> *Response:* For data-scarce domains, the transferability metrics usually do not work well (LEEP scores vary a lot with data size). To tackle that and put more weights to the better DAFT models, we used LR based method. Depending on the performance of the DAFT models, the LR method puts more weights to the better performing models. More on the weight learning in Appendix A.6.1.
>
>
> **W5:** "Given that if ..."
>
> *Response:* Depending on the training data size and models structure, ensemble of thousands of models can be more costly compared to fine-tuning the model. However, we want to emphasize that with DAFT-E, we only need to perform forward pass once, while full fine-tuning may need multiple forward and backward passes. Also, if we have a small amount of data from the target domain, we can use LR or similar methods to filter better DAFT models.

---

> > ### Author Response · Authors · 2025-03-25
> > **Kind Reminder: Response to Reviewer Comments Submitted and Changes Made in Manuscript**
> >
> > Dear Reviewer, we have incorporated the changes that you suggested in the main manuscript and have uploaded it. Please let us know if you are satisfied with the modifications or have more concerns. The description on the changes suggested by you are as follows:
> >
> > 1.	We have searched for sentiment analysis models from Huggingface and sorted the models in terms of their popularity in Huggingface. Currently, we have selected the top 25 models that can understand English and checked the performance of DAFT-E$^z$. The results looked very promising, and now we are adding more models to the DAFT model set (some filtering is needed due to other language models and output mapping). We plan to add 100 models and report the results.
> > 2.	A line is added on the main text about learning strategy. (pages 6,7)
> > 3.	We are checking Sof-IoU-EEP and hope to report some results within this discussion/revision timeframe. However, please note that our focus in this work is not to find the best option to measure the transferability of DAFT models, but rather to show that the publicly available models have transferability to different tasks.
> > 4.	We have added more discussion and definition of the DAFT, and measurement of the similarity between tasks. (page 4)
> > 5.	We have added a line in page 1. Also, we have read the rest of the paper to check for consistency issues.
> > 6.	We have extended the *Broader Impact Statement* by adding the suggested texts.
> >
> > Please note that we still have some pending changes, and we will notify you again when we have a new revision with those changes incorporated.
> >
> > Thanks
> > -Authors

---

### Review · Reviewer_8MW6 · 2025-03-11

**Summary Of Contributions:**

This paper proposes using existing domain or task adjacent fine-tuned models (DAFT models) for zero-shot and few-shot learning under the scenario of data-scarce tasks. Specifically, the authors have evaluated five different approaches to leverage the DAFT models, such as fine-tuning DAFT models (DA(FT)^2) and ensemble DAFT models (DAFT-E).

The key contributions are:

1. The authors evaluate five approaches with respect to fine-tuning and ensembing DAFT models. It is a solid evaluation.
2. The authors provide mathematical proofs to show that ensembling DAFT models are better than randomly selecting a single DAFT model for data-scare tasks.

**Audience:**

Yes

**Broader Impact Concerns:**

I don't have any broader impact concerns for this paper.

**Claims And Evidence:**

Yes

**Requested Changes:**

1. Add the comparison to LLM-Blender and more LLM ensemble methods to Table 1.

2. For Figure 6, the experimental results are not consistent across different datasets. For example, DAFT-E can surpass Best DAFT on Amazon even with limited training data size, while Best DAFT is always better than DAFT-E on IMDB. Could the authors provide any explanations and further analysis about it? Same thing happened in Figure 7 as well.

3. The caption of Figure 4 confuses me. BART-LARGE-MNLI and ROBERTA-LARGE-MNLI are classification-based models which does not have any emergent abilities such as in-context learning and instruction-following. What does ‘prompt on larger LLMs’ mean?

4. For Table 2, I suggest making it a line plot instead of a table, thus we can observe the tendency with respect to the number of shots easily.

5. It is better to unify the notations for referred figures (e.g., Figure 2 -> Fig 2).

6. In Figure 5, for glue-qqp, what are the DAFT FMs that can perform better than DAFT-E^Z (points larger than zero)? Please analyze the reason for this phenomenon.

**Strengths And Weaknesses:**

Advantages:

1. To solve data-scarce tasks, the paper explores a lightweight and scalable approach by utilizing available DAFT models. It provides a new view to solve data-scarce tasks.
2. Compared to Model Soup, DAFT-E and DAFT-E^{Z} can handle the case of multi-architecture ensemble (Table 1).
The methods are evaluated on six different datasets, which is very sufficient.

Disadvantages:

1. For the comparison to Model Soup (Table 5 and 6). The performance gain is quite limited (no more than 1%).

2. The performance of DAFT models depends heavily on their similarity to the target task. This paper does not propose a method for selecting the DAFT models from a more general model set.

---

> ### Author Response · Authors · 2025-03-18
> **Response to Requested Changes and Weaknesses**
>
> ### Dear Reviewer, thank you for your insightful comments and suggestions. We have made every effort to address them thoroughly as follows. We kindly invite your feedback on our responses.
> ### **Requested Changes**
> **RC1:** "Add the comparison..."
>
> *Response:* Thank you for the suggestion. We will add LLM-Blender [1] and AdaMerging [2] to the methods in Table 1.
>
> [1] Jiang et al., Llm-blender: Ensembling large language models with pairwise ranking and generative fusion. arXiv preprint arXiv:2306.02561 (2023).
>
> [2] Yang et al., AdaMerging: Adaptive Model Merging for Multi-Task Learning, ICLR, 2024.
>
> **RC2:** "For Figure 6, ..."
>
> *Response:* From the trend observed in our evaluation, we found that in most cases DAFT-E outperforms the single best DAFT with few-shot weight training. However, there can be cases where a smaller number of examples is not good enough to adjust the weights of the DAFT models to outperform the single best DAFT. For the case of IMDB that is the case, and with more examples from the target domain, we have observed that DAFT-E reaches the single best DAFT. As an example, for the case of SST2, we observe the
> performance of DAFT-E becomes comparable to single best DAFT with n = 128.
>
> **RC3:** "The caption of Figure 4 ..."
>
> *Response:* Thank you for pointing this out. When we downloaded those models from Huggingface, all three (large) models needed some sort of prompting/output mapping, and hence we used the term ‘prompt on larger LLMs’. Currently, BART-LARGE-MNLI and ROBERTA-LARGE-MNLI are fully classification-based models and we will change that in the text accordingly.
>
>
> **RC4:** "For Table 2, ..."
>
> *Response:* Thank you for this suggestion. We have generated the line plot and will add it beside the current table.
>
> **RC5:** "It is better ..."
>
> *Response:* Thank you for noticing that, we will do a proofread and modify those instances.
>
> **RC6:** "In Figure 5, ..."
>
> *Response:* Yes, that is the case: points larger than zero are performing better than DAFT-Ez. Although such cases are infrequent, for this specific case of glue-QQP, DAFT-Ez did not perform well. The main reason behind this is the poor performance of the DAFT models for the glue-qqp (as can be seen in Fig. 2b-right, the performance of the DAFT models are poor compared to FFT, i.e., (2,2) position of the matrix). Moreover, we observed a (somewhat) similar issue (but not as prominent) for the tweet dataset.
>
> On the other hand, please note that the numerical values of the relative improvement, which are (1 floating point): -9.7, -7.8, 0.3, 1, 1.2, and 10, are negative (-ve) on the average, which follows our claim that ensembling of DAFT models is better than randomly selecting a single DAFT model for data-scare tasks.
>
> ### **Weakness**
> **W1:** "For the comparison..."
>
> *Response:* It is true that the performance gain is not prominent compared to Model soup, but the use of DAFT-E has some other advantages over Model Soup as discussed in Related Work (Paragraph 4 and Table 1).
>
> **W2:** "The performance of DAFT ..."
>
> *Response:* From the manuscript: when a model is fine-tuned with a dataset that has the same task criterion as the target data, we call it a DAFT for the target task. While this broad-brushed selection method may seem crude (and some of the DAFT models selected may be under-performing), this is driven by practical considerations. When we do not have any training data from the target task, selecting DAFT models based on more quantitative criteria is difficult. Lastly, when we have some limited data from target task for training, the LR method proposed here can select the DAFT models. Some other potential methods are LEEP, Cosine-similarity etc.

---

> > ### Author Response · Authors · 2025-03-25
> > **Kind Reminder: Response to Reviewer Comments Submitted and Changes Made in Manuscript**
> >
> > Dear Reviewer, we have incorporated the changes that you suggested in the main manuscript and have uploaded it. Please let us know if you are satisfied with the modifications or have more concerns. The description on the changes suggested by you are as follows:
> >
> > 1.	LLM-Blender and Adamerging are added for comparison in Table 1. (page 2)
> > 2.	We have commented on this requested change (please see our previous response); if you think that the comment addresses your concern, we will add it to the manuscript.
> > 3.	BART-LARGE-MNLI, ROBERTA-LARGE – type is modified from prompt to classification-based. (pages 5, 6)
> > 4.	Table 2 was converted to line-plot and is added on the side of the table. (page 7)
> > 5.	All Figure references are now denoted with Fig.
> > 6.	We have commented on this requested change (please see our previous response); if you think that the comment addresses your concern, we will add it to the manuscript.
> >
> > Please note that we still have some pending changes and we will notify you again when we have a new revision with those changes incorporated.
> >
> > Thanks
> > -Authors

---

### Review · Reviewer_Ccad · 2025-03-14

**Summary Of Contributions:**

This paper presents a relevant investigation into leveraging the growing number of publicly available fine-tuned LLMs termed Domain Adjacent Fine-Tuned (DAFT) models,  to address domains with scarce training data.
The authors explore several strategies, including zero-shot inference with individual DAFT models (DAFT Z), ensembling multiple DAFT models in a zero-shot manner, fine-tuning a single DAFT model, and ensembling DAFT models with lightweight few-shot adaptation of ensemble weights.

**Audience:**

Yes

**Claims And Evidence:**

Yes

**Requested Changes:**

1. The definition of "base model" in Section 1 as a pre-trained LLM with an untuned task-specific header layer contrasts with the later description of creating DAFT models by fine-tuning these base models (including the header layer) on specific datasets. Clarifying the stage at which the header layer is introduced and tuned would improve consistency.

2. Table 2 indicates that DA(FT)2 generally outperforms FT in few-shot settings. However, the trend of decreasing average relative improvement with increasing numbers of shots (n) across most datasets suggests that the initial advantage of starting with a DAFT model for fine-tuning diminishes as more target data becomes available. This trend and the point at which fine-tuning a base model might become comparable to or better than fine-tuning a DAFT model could be discussed more explicitly.

3. The theoretical analysis in Section 5 relies on the assumption of a convex loss function and introduces a distance measure $\rho(D_{\nu(i)}, D_T)$ between datasets. However, the specific definition and practical computation of this distance measure are not provided

**Strengths And Weaknesses:**

Strengths:

- The paper addresses a key challenge in modern LLM research—how to leverage the ever-growing pool of fine-tuned models for applications where training data is limited.
- The paper examines multiple approaches: zero-shot and few-shot methods and include empirical comparisons and theoretical guarantees that support the ensemble methods  (Propositions 1 and 2) .
- By emphasizing computational cost (e.g., no trainind required for DAFT-Z/E methods) and ease of integration (using models available via APIs), the paper is practically appealing for low-resource settings.

Weaknesses:

1. A central assumption is that there is a sufficiently diverse and high-quality pool of fine-tuned models available. the authors acknowledges that if none of the available models are truly adjacent to the target domain, both the ensemble methods and even the best single DAFT may underperform. This reliance on the availability of “good” DAFTs is a potential weakness that might limit the generalizability of the approach to less-studied tasks.

2. The empirical evaluation primarily focuses on sentiment analysis and textual similarity tasks. The transferability of the findings and the effectiveness of the proposed methods for a broader range of NLP tasks. This is especially concerning since recent autogressive models eg., Qwen2.5 perform very well on these tasks. Evaluating question answering and text generation remain unexplored.

3. The performance on different datasets (e.g., sentiment analysis and textual similarity tasks) with some datasets (like Tweet and Cornell) showing peculiar behavior. E.g: , the counterintuitive finding that DAFT-SST2 sometimes underperforms compared to DAFT-Cornell highlights potential inconsistencies in how domain similarity is assessed.

4. In the theoretical analysis (particularly in Proposition 2 and Corollary 1), the paper assumes that all base models perform similarly. In practice, however, models like Roberta, BERT, and XLNet may behave differently, especially across diverse domains. This simplification might limit the applicability of the theoretical guarantees when the underlying models have markedly different capacities or inductive biases.

---

> ### Author Response · Authors · 2025-03-18
> **Response to Requested Changes and Weaknesses**
>
> ### Dear Reviewer, thank you for your insightful comments and suggestions. We have made every effort to address them thoroughly as follows. We kindly invite your feedback on our responses.
> ### **Requested Changes**
> **RC1:** "The definition of..."
>
> *Response:* Thank you for your suggestion. We plan to define the base model and the DAFT model for some specific task as separate definitions at the beginning of Section 3 for further clarity. In particular, we plan to add the following:
>
> Base Model: a model that has a pre-trained LLM and a task-specific (un-tuned) header layer added on top of the LLM.
>
> A base model’s header layer can vary depending on the task, and it can be a domain adjacent fine-tuned model (DAFT) for a task when the base model is fine-tuned (either the header layer or the full model) with a dataset that has the same task criterion as the target data. We define the DAFT model as follows:
>
> DAFT: A domain adjacent fine-tuned model (DAFT) is a fine-tuned base model, if the model can be used for the same (general) task as the target task.
>
> **RC2:** "Table 2 indicates..."
>
> *Response:* Thank you for your suggestion, and we agree with it. We are planning to add a figure alongside the current Table 2 to show how FT catches up to DA(FT)$^2$ as the number of samples for fine-tuning increases. We will also add a line saying: “Please note that if there are a moderate amount of data-samples (for the current example, n>128) available from the target domain (not a data-scarce environment) FT is expected to perform similarly as DA(FT)$^2$.”
>
> **RC3:** "The theoretical analysis..."
>
> *Response:* Thank you for pointing this out. We assume a convex loss function since the loss is calculated using the output probabilities of each DAFT model, which is a continuous value. Also, the popular loss functions like MAE, or MSE are directly proportional to the distance between $y_{pred}$ and $y_{true}$, and are convex functions [1]. On the other hand, some practical computations of dataset distance can be the Wasserstein distance metric [2] or Jensen-Shannon Divergence (JSD). We will add discussions on these in our manuscript.
>
> [1] Terven et al., Loss functions and metrics in deep learning. arXiv preprint arXiv: 2307.02694(2023).
>
> [2] Panaretos et al., Statistical aspects of Wasserstein distances. Annual review of statistics and its application 6, no. 1(2019) : 405 − 431.
>
> ### **Weakness**
> **W1:** "A central assumption..."
>
> *Response:* It is true that the quality and relevance of the available DAFT models affect the final performance (which is also what our theoretical analyses show). Thus, favorable results require “good” DAFT models. However, we would like to reiterate that the motivating question for our work here is “how can we use the plethora of fine-tuned LLMs for a data-scarce task in a computationally efficient manner?” To this end, we do not claim that we can get the best performance but rather try to leverage what is available.
>
> **W2:** "The empirical evaluation..."
>
> *Response:* We do consider only two classes of tasks, and do not consider other applications such as question answering and text generation. However, we do thoroughly evaluate various datasets and scenarios for these two classes. It is true that there are many general-purpose models out there that can simultaneously perform well on multiple classes of tasks. However, it is important to recognize that these models are pre-trained on a large corpus, and subsequently fine-tuned on various auxiliary tasks. Our focus in this paper is on leveraging already openly available fine-tuned models to solve data-scarce tasks in a compute-efficient manner rather than training a single general-purpose model.
>
> **W3:** "The performance..."
>
> *Response:* We do not think this inconsistency is due to domain similarity assessment. From our study we found that the phenomenon occurred due to the test dataset of SST2 being more aligned with the Cornell dataset (train & test). From the perspective of the real world, this is an important observation. It presents a scenario in which the test data from a target domain is significantly more aligned to some DAFT model training data rather than the target domain training data. For these special cases, DAFT models might be a great solution that outperforms even the FFT model.
>
> **W4:** "In the theoretical..."
>
> *Response:* Please note that we do not generally assume that ‘all base models perform similarly’. Our comment is a bit more specific (see the line before Corollary 1) – we state that ‘Usually base models (derived from FMs which are all quite large) perform similarly when finetuned on the full target dataset’. This is not claiming that they have the same performance across diverse domains, but rather that, upon **full fine-tuning with a large enough target task data**, the fine-tuned models (derived from different base models) can achieve quite high performance on the target task, which is a reasonable assumption in our opinion.

---

> ### Author Response · Authors · 2025-03-25
> **Kind Reminder: Response to Reviewer Comments Submitted and Changes Made in Manuscript**
>
> Dear Reviewer, we have incorporated the changes that you suggested in the main manuscript and have uploaded it. Please let us know if you are satisfied with the modifications or have more concerns. The description on the changes suggested by you are as follows:
>
> 1.	More discussion and definition of the DAFT, and measurement of the similarity between tasks. (page 4)
> 2.	Table 2 was converted to line-plot and is added on the side of the table. (page 7)
> 3.	The reasoning behind the assumption of convex loss function and practical dataset distance measurement metric (page 12 -footnote and page 18).
>
> Also, please check our previous responses to the requested changes and weaknesses. Please note that we still have some pending changes and we will notify you again when we have a new revision with those changes incorporated.
>
> Thanks
> -Authors

---

### Author Response · Authors · 2025-03-28
**All revisions requested by the reviewers have been incorporated into the manuscript.**

### **All revisions requested by the reviewers have been incorporated into the manuscript with new texts and figures; the new texts are colored in MidnightBlue. The major changes in the manuscript are as follows:**

1. We have added DAFT-E$^z$ and DAFT-E performance comparison with DAFT models downloaded directly from online public repositories. (pages 4, 18, and 19; figures 17, 18, and 19)
2. Added LLM-Blender and Adamerging for comparison in Table 1. (page 2)
3. Table 2 was converted to line-plot and is added on the side of the table. (page 7)
4. Discussion on some more related work are added. (pages 2, 3)
5. Discussed additional transferability metrics, e.g., SoftIoU-EEP, that can be used in DAFT-E to get the top subset(s) of DAFT models. (page 9)
6. BART-LARGE-MNLI, ROBERTA-LARGE – type is modified from prompt to classification-based. (pages 5, 6)
7. More discussion and definition of the DAFT, and measurement of the similarity between tasks. (page 4)
8. Modification of the broader impact statement. (page 13)
9. A sentence is added on the main text about learning strategy. (pages 6, 7)
10. The reasoning behind the assumption of convex loss function and practical dataset distance measurement metric (page 12 -footnote and page 18).
11. Some modifications in writing / typos pointed out by the reviewers. (pages 1, 6, 15, 16)

---

### Decision · Action_Editor_ds1H · 2025-04-24

**Recommendation:** Accept with minor revision

**Comment:**

The paper studies an interesting problem setting on how to leverage fine-tuned LLMs for specific applications without requiring extensive task-specific/customized fine-tuning. The authors propose several intuitive methods for zero-shot and few-shot learning of DAFT models, and demonstrate the effectiveness on a set of classification benchmarks. The reviewers generally appreciated the important and interesting problem being studied, and the empirical effectiveness of the methods. They also raised several suggestions, and the authors have mostly addressed them by adding comparisons with ensembling methods and providing clarification regarding the technical details.
In the final recommendation, one remaining concern from the reviewers is that "the utility of the proposed methods is questionable because of the zero-shot performance of recent LLM on all the benchmarks tested in this paper, which removes the need of doing such technique." I agree that the benchmarks used in this submission are relatively trivial and outdated for evaluating LLMs, especially when the authors consider data scarcity settings (which naturally entails the assumption that the target task is relatively challenging). Therefore, the authors should provide more evaluation on at least a few modern LLM benchmarks (e.g., MMLU, MATH, GPQA) in the revision.

**Audience:**

Yes

**Claims And Evidence:**

The claims are mostly supported. One limitation is that the type of tasks studied in this paper is mostly on simple binary classification tasks, which are known to be rather trivial for LLMs. As the goal of the paper is "we explore different utilization techniques of these existing DAFT models for data-scarce problems, i.e., tasks for which data is not available or limited", the authors should explore more challenging tasks where the data scarcity issue is more pronounced (e.g., MMLU, math reasoning, or hard QA).

---

> ### Author Response · Authors · 2025-05-24
> **Minor Revision Addressed**
>
> Dear Action Editor,
>
> We have revised the manuscript to address the minor revision requested by you. The specific modifications are as follows:
>
> 1. A new figure and accompanying text have been added to illustrate and discuss the performance of DAFT models on 9 medical-related datasets from MMLU (see Page 6, last paragraph, and Figure 4).
>
> 2. Figure 7 and the corresponding discussion on Page 9 present a comparison of the performance of $DAFT\text{-}E^z$ against the individual DAFT models on these 9 datasets.
>
> 3. The potential advantages of employing $DAFT\text{-}E$ for the MMLU datasets are analyzed on Pages 11–12, supported by the results shown in Figure 10.
>
> Please let us know if any additional revisions are required.
>
> Thanks,
> -Authors

---

> > ### Comment · Action_Editor_ds1H · 2025-05-25
> > **Thanks for the revision**
> >
> > Dear Authors,
> >
> > Thank you very much for submitting the revision. I think the newly added results are helpful to strengthen the contribution of the paper.
> >
> > One minor note: Could you please add the actual OpenReview link under the author block (currently it's still a placeholder) "Reviewed on OpenReview:"
> >
> > I'll approve the camera-ready version once this is fixed.

---

> > > ### Author Response · Authors · 2025-05-25
> > > **OpenReview link added**
> > >
> > > Dear Action Editor,
> > > We’re glad that the new results have addressed your concerns and agree that they strengthen the paper. We have modified the OpenReview link in the pdf. Please approve the camera-ready version at your convenience.
> > >
> > > Thanks,
> > > -Authors.